# The Puy de Dôme ICe Nucleation Intercomparison Campaign (PICNIC): Comparison between online and offline methods in ambient air

Larissa Lacher[1], Michael P. Adams[2], Kevin Barry[3], Barbara Bertozzi[1], Heinz Bingemer[4], Cristian Boffo[5], Yannick Bras[6], Nicole Büttner[1], Dimitri Castarede[7], Daniel J. Cziczo[8,9], Paul J. DeMott[3], Romy Fösig[1], Meghan Goodell[8], Kristina Höhler[1], Thomas C. J. Hill[3], Conrad Jentzsch[10], Luis A. Ladino[11], Ezra J. T. Levin[3], Stephan Mertes[10], Ottmar Möhler[1], Kathryn A. Moore[3], Benjamin J. Murray[2], Jens Nadolny[1], Tatjana Pfeuffer[5], David Picard[6], Carolina Ramírez-Romero[11], Mickael Ribeiro[6], Sarah Richter[4], Jann Schrod[4], Karine Sellegri[6], Frank Stratmann[10], Benjamin E. Swanson[3], Erik S. Thomson[7], Heike Wex[10], Martin J. Wolf[8], and Evelyn Freney[6]

[1]Institute of Meteorology and Climate Research, Karlsruhe Institute of Technology, Karlsruhe, 76021, Germany
[2]School of Earth and Environment, University of Leeds, Leeds LS2 9JT, United Kingdom
[3]Department of Atmospheric Science, Colorado State University, Fort Collins, 80523, Colorado, United States
[4]Institute for Atmospheric and Environmental Sciences, Goethe University Frankfurt, Frankfurt, 60438, Germany
[5]Bilfinger Noell GmbH, Würzburg, 97080, Germany
[6]Laboratoire de Météorologie Physique, Université Clermont Auvergne, 63178 Clermont-Ferrand, France
[7]Department of Chemistry and Molecular Biology, University of Gothenburg, Gothenburg, 40530, Sweden
[8]Department of Earth, Atmospheric and Planetary Sciences, Massachusetts Institute of Technology, Cambridge, 02139, Massachusetts, United States
[9]Department of Earth, Atmospheric, and Planetary Sciences, Purdue University, West Lafayette, 47907, Indiana, United States
[10]Leibniz Institute for Tropospheric Research, Leipzig, 04318, Germany
[11]Institute for Atmospheric Sciences and Climate Change, Universidad Nacional Autónoma de México, Mexico City, 04510, Mexico

*Correspondence to*: Larissa Lacher (larissa.lacher@kit.edu)

**Abstract.**

The formation of ice crystals in clouds is initiated by specific aerosol particles, termed ice-nucleating particles (INPs). Only a tiny fraction of all aerosol particles are INPs and their concentration over the relevant temperature range for mixed-phase clouds (< -38 °C) covers up to ten orders of magnitude, providing a challenge for contemporary INP measurement techniques. INP concentrations can be detected online with high-time resolutions of minutes, or offline, where aerosols are collected on filters for hours to days. Here we present measurements of INP concentrations in ambient air under conditions relevant to mixed-phase clouds from a total of ten INP methods over two weeks in October 2018 at the Puy de Dôme observatory in central France. INP concentrations were detected in the immersion freezing mode, between ~ -5 °C and -30 °C. Two continuous flow diffusion chambers (CFDC; Colorado State University-Continuous Flow Diffusion Chamber, CSU-CFDC; Spectrometer for Ice Nuclei, SPIN) and an expansion chamber (Portable Ice Nucleation Experiment, PINE) measured the INP concentration with a time resolution of several minutes and at temperatures below -20 °C. Seven offline freezing techniques determined the temperature-dependent INP concentration above ~ -30 °C using water suspensions of filter-collected particles sampled over 8 hours (FRankfurt Ice Nuclei Deposition FreezinG Experiment, FRIDGE; Ice Nucleation Droplet Array, INDA; Ice Nucleation Spectrometer of the Karlsruhe Institute of Technology, INSEKT; Ice Spectrometer, IS; Leipzig Ice Nucleation Array, LINA; LED based Ice Nucleation Detection Apparatus LINDA; Micro-Orifice Uniform Deposit Impactor–Droplet Freezing Technique, UNAM-MOUDI-DFT). A special focus in this intercomparison campaign was placed on having overlapping sampling periods for the methods: INP concentrations measured with the online instruments were compared within 10 minutes and at the same temperature (±1 °C), while the filter collections for offline methods were started and stopped simultaneously and the obtained INP freezing spectra were compared at 1 °C steps. The majority of INP concentrations measured with PINE agreed well with the CSU-CFDC within a factor of two and five (71% and 100% of the data, respectively). There was a consistent observation of lower INP concentration with SPIN, and only 35% of the data are within a factor of two from the CSU-CFDC, but 80% of the data are still within a factor of five. This might have been caused by an incomplete exposure of all aerosol particles to water-supersaturated conditions within the instrument – a feature inherent to CFDC-style instruments – demonstrating the need to account for aerosol lamina spreading when interpreting INP concentration data from online instruments.

The comparison of the offline methods revealed that more than 45% of the data fall within a factor of two from the results obtained with INSEKT. Measurements using different filter materials and filter holders revealed no difference in the temperature-dependent INP concentration at overlapping temperatures. However, consistently higher INP concentrations were observed from aerosol filters collected on the rooftop at the Puy de Dôme station without the use of an inlet, compared to measurements performed simultaneously behind the whole air inlet system.

## 1   Introduction

The first formation of ice in mixed-phase clouds is triggered by specific aerosol particles, called ice-nucleating particles (INPs; Vali et al., 2015). The presence of INPs is important for the formation and further development of clouds since they can determine cloud phase (e.g., by a rapid cloud glaciation and associated dissipation effect; Campbell and Shiobara, 2008; Murray et al., 2012; Paukert and Hoose, 2014; Kalesse et al., 2016; Desai et al., 2019; Murray and Liu, 2022; Carlsen and David, 2022; Creamean et al., 2022; Sze et al., 2023) and related radiative

properties (e.g., Vergara-Temprado et al., 2018). In addition, INPs have an impact on precipitation formation (e.g., Mülmenstädt et al., 2015; Field and Heymsfield, 2015; Fan et al., 2017). However, the identification and quantification of ambient INPs remain challenging due to their rarity (e.g., Kanji et al., 2017) and limitations in measurement techniques (DeMott et al., 2017; Cziczo et al., 2017).

Different methods to quantify ambient INP concentrations exist and are categorized into online instruments and offline freezing techniques. Online instruments measure real-time INP concentrations with a high temporal resolution (seconds to minutes). It has been shown that INP concentration can fluctuate considerably within short sampling times of minutes and hours (e.g., Prenni et al., 2009; Lacher et al., 2017; Welti et al., 2018; Paramonov et al., 2020). Therefore, online methods are required to catch such variability, and relate it to, e.g., changes in air

mass and aerosol properties. On the other hand, currently available online instruments for ambient measurements typically sample only a few litres of air per minute. This limits the ability of these methods to detect low INP concentrations in ambient air. Offline methods are based on collecting aerosol particles on sampling substrates or into liquids, typically over longer periods of hours to days, and therefore can collect larger volumes of air ($\sim$ m$^3$), increasing the likelihood of sampling the very rare INPs active at the highest temperatures. Results from offline

INP measurements can also be obtained for shorter periods, however, this impacts the limit of detection and may lead to a lower or even zero number of very rare INPs. Due to the labor-intensive filter collection and analysis procedures, online methods are often preferred to measure INPs with a high-time resolution. While offline INP analysis could impact the properties of the collected INPs due to the sampling and analysis procedure (e.g., physical or chemical alteration, particle breakup, loss of coating material), they also allow for special treatments,

for example investigating the contribution of organic INPs by heat or peroxide treatments (e.g., Hill et al., 2016), to improve our understanding of the INP properties.

In order to accurately quantify INPs, existing methods need to be validated and compared with each other, to address potential systematic biases. A set of different methods were compared in laboratory studies using well-known aerosol particles, e.g., by sharing samples of SNOMAX®, cellulose, or illite-rich samples amongst the

community of the Ice Nuclei Research Unit (INUIT; Wex et al., 2015; Hiranuma et al., 2015; Hiranuma et al., 2019), during the Leipzig Ice Nucleation chamber Comparison (LINC; Burkert-Kohn et al., 2017), and during the Fifth International Workshop on Ice Nucleation phase 2 (FIN-02; DeMott et al., 2018). Those experiments revealed a generally good agreement among a large set of freezing methods. Hiranuma et al. (2015) indicated that the aerosol particle generation method (dry versus wet suspension) can lead to changes in detected INP concentrations,

which was also found by other laboratory studies (Emersic et al., 2016; Boose et al., 2016b). Moreover, in these studies, it was shown that the methods' comparability depended on the chosen aerosol particle type and nucleation temperature: Below -10 °C, instruments showed good agreement using SNOMAX® and natural dust samples. Discrepancies occurred using SNOMAX® above -10 °C, with illite NX above -25 °C, and with potassium feldspar between -20 and -25 °C.

Another aspect that is crucial for the intercomparison of ice nucleation techniques is the size range of aerosol particles that are INPs. Typically, online instruments, such as continuous flow diffusion chambers (CFDCs), limit the aerosol sampling size to diameters below $\sim$ 3 µm (e.g., Rogers et al., 2001), as they commonly aim at measuring freshly formed ice crystals within the chamber using optical particle counters (OPCs), and a size overlap with unactivated large aerosol particles must be avoided because optical size alone is often the basis for distinguishing

frozen and unfrozen particles. By contrast, offline techniques are able to sample those larger aerosol particles, e.g., using inline or open-faced filter holders. Many of these techniques collect aerosol particles on filters, which could lead to a reduced sampling of particles smaller than the pore size. However, theoretical calculations indicate that most particles smaller than the pore size will be sampled (Spurny and Lodge, 1972), and in a study by Ogura et al. (2016) it was found that ~70% of particles smaller than 100 nm are collected on the direct surface of 200 nm

Nuclepore filters. Also, Soo et al. (2016) report that filters can have a high collection efficiency for particles much smaller than their nominal pore size, dependent on filter material and sampling flow. Moreover, not all particles may be released completely from the filter during the washing-off procedure before analysis, and particle collection efficiency can also be reduced by a possible bounce from the collection substrate when using stage impactors. Thus, the role of the dominant size of INPs is an important aspect in assessing the suitability of an INP

method to capture the picture of ambient conditions. Super-micrometer particles have been found to contribute to the majority of INPs in different studies in North America and Europe (Mason et al., 2016), the Arctic (Creamean et al., 2018), Cabo Verde islands (Gong et al., 2020), and the Yucatan Peninsula (Córdoba et al., 2021), however, with a varying fraction, potentially depending on the sampling location, the aerosol type, and the nucleation temperature. Contrastingly, the analysis of ice crystal residuals in the lower free troposphere revealed that the

majority of particles were submicron in size (e.g., Mertes et al., 2007; Schmidt et al., 2017). Ice-active organic particles from marine sources were found to be submicrometer (Wilson et al., 2015) and supermicrometer (Mitts et al., 2021) in size, and it is unclear which size range is dominating the INP population in such remote marine environments. In laboratory-based intercomparison studies, it was suggested that generally good agreement between methods was achieved by controlling the aerosol particle size distributions used for the INP experiments

(Wex et al., 2015; Burkert-Kohn et al., 2017 DeMott et al., 2018). At ambient conditions, however, aerosol particles and INPs can span a wide size range, which can be crucial for determining the real ambient INP concentration, and for comparing INP measurement techniques that cover different size ranges (Knopf et al., 2018). This may be especially relevant for mineral dust, acknowledged to be a key INP in the troposphere at temperatures below -15 °C (e.g., Atkinson et al., 2013). The occurrence of supermicrometer dust particles close to

emission sources is certainly higher compared to locations further away.

    Ambient INP concentrations can be close to typical instrument detection limits (Boose et al., 2016a) and the way measurements close to detection limits are considered for averaging INP concentration over longer sampling intervals, which can be done for comparing different instruments, is another important aspect of making ambient measurements. Ambient INPs show a wide range of concentration across the relevant temperature range (e.g.,

Kanji et al., 2017), and it should be ensured that even low numbers of INPs, close to instruments' detection limits, are captured.

    By conducting measurements on ambient aerosols, impacts from aerosol generation methods and domination by a single INP type are avoided, and the instruments are compared under realistic conditions such as the naturally low INP number concentration. DeMott et al. (2017) presented a field-based intercomparison campaign using four

offline techniques and an online instrument (Colorado State University Continuous Flow Diffusion Chamber CSU-CFDC) at different locations in the Western USA, including agricultural areas, mountainous desert regions, and a coastal site. They generally found good agreement between instruments, especially when measurements were performed synchronously. However, a high bias for offline methods, sampling particles onto filters or into a bulk liquid, against an online method was observed below -20 °C. It is unclear if this might have been caused by a

breakup of aggregates by partial solvation of aerosols that contain more than one INP, or if larger INPs were not captured by the online method used in that study. In a recent study by Brasseur et al. (2022) in the Finnish boreal forest, three online instruments were compared over four days at a nucleation temperature below -29 °C and generally showed a good agreement. Such intercomparison efforts need to be expanded to cover the full range of mixed-phase cloud temperatures, and conducted in environments in which mixed-phase clouds occur. INP
intercomparison activities are especially relevant due to ongoing efforts for the establishment of INP monitoring networks. For example, at the European level, the ACTRIS (Aerosol, Clouds and Trace Gases Research Infrastructure) Topical Centre for Cloud In Situ measurements is currently in an implementation phase to include INP concentration as a parameter to be monitored at specific research stations. For such an effort, it is crucial to ensure that INP concentrations are accurately quantified using different online or offline instruments. This will
contribute to developing harmonized data sets.

Here we present results from the Puy de Dôme ICe Nucleation Intercomparison Campaign (PICNIC). The Puy de Dôme station is a mountaintop station situated in central France at an altitude of 1465 m above sea level. Given its altitude, it is often affected by air masses transported over long distances and, hence, can contain aerosol particles emitted from source regions far away. It is also an environment in which clouds form and occur, thus the
aerosol population being present at the Puy de Dôme is relevant for aerosol-cloud interactions. During PICNIC, seven offline techniques and three online instruments were compared over 14 days in October 2018. The aim here was to test the measurement techniques against each other in their original operation configuration, as each of them are well-established methods and were used already in several campaigns, and we wanted to create a link between these activities, without changing measurement protocols. A key aspect is that offline and online
instruments were intercompared during the same filter sampling time (offline instruments) or within 10 minutes (online instruments). Only when intercomparing the online to the offline methods, the time intervals were not perfectly overlapping. Moreover, two main sampling locations inside the laboratory, via a total aerosol inlet, and one location directly outside on the laboratory´s rooftop were used, addressing potential sampling biases due to particle losses in the inlet and by the use of upstream impactors necessary for some online instruments. Advances
over past studies come from the use of a larger suite of methods and coordination of longer and shared sampling times.

## 2 Methods

### 2.1 Measurement location and time

The PICNIC campaign took place from the 7th to the 20th of October 2018, at the Puy de Dôme (1465 m above sea
level), which is located in central France. An overview of the measurement campaign will be presented by Freney et al. (in preparation), and some details are given by Bras et al. (2022). The station is located on a mountain chain, thus the site is suited to sample atmospheric layers originating in the boundary layer, as well as in the lower free troposphere (Asmi et al., 2012; Farah et al., 2018; Baray et al., 2020). The site is operated by the *Observatoire du Physique du Globe de Clermont Ferrand* (OPGC) and run by the *Laboratoire de Météorologique Physique* (LaMP)
and is an observational facility of ACTRIS and the Global Atmospheric Watch measurement programs. Continuous measurements of meteorological conditions, as well as aerosol physical and chemical properties, are provided. The submicrometer aerosol particle size distribution was measured using a custom-made scanning mobility particle sizer (with a particle diameter range from 10 – 560 nm) operated with a condensation particles

counter (CPC, model 3010, TSI) via a whole air inlet (WAI) with a 50% cut-size diameter of 30 μm. Also, aerosol particle concentrations between 0.5 and 2.5 μm were sporadically derived from the OPC of the CSU-CFDC (see section 2.2.1) and corrected for a growth factor based on an assumption of ammonium sulphate composition.

Moreover, the transmission efficiency of the WAI is dependent on wind speed. Calculations show that at values of 7 (10) m s$^{-1}$, 93% (84%) of the particles with a diameter of 10 μm are entering the inlet (Hangal and Willeke, 1990; Baron and Willeke, 2002). INP concentration measurements were conducted inside the laboratory, via two identical WAIs, as well as on the rooftop (Fig. 1). Full details on the measurement setup of all online and offline techniques are provided in the following sections.

In this study, we consider an agreement of INP concentration measurements if observations are consistent within factors of 2 and 5. It was indicated that the representations of INPs in models need to be predictable within a factor of 10 to not change cloud microphysics (Phillips et al., 2003), and our chosen values of 2 and 5 are thus even more conservative and can be considered to represent a good (factor 2) and agreeable (factor 5) comparison.

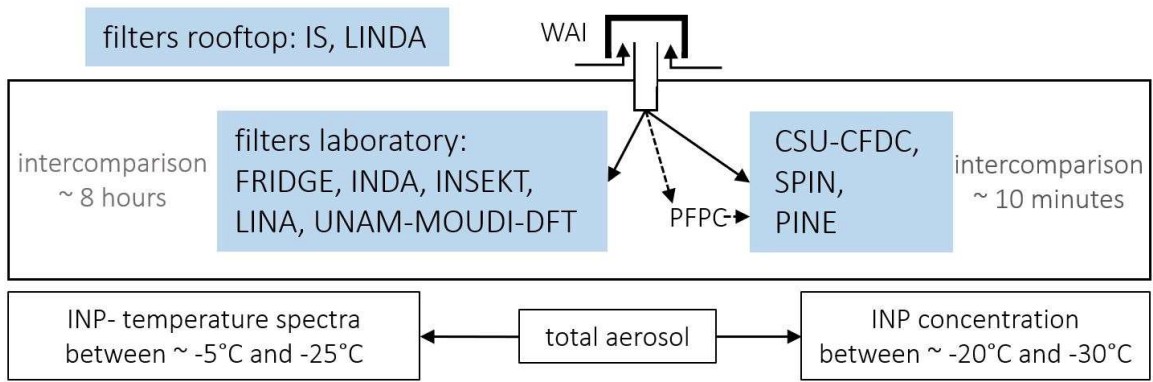

Figure 1: Setup of online instruments CSU-CFDC (Colorado State University-Continuous Flow Diffusion Chamber, SPIN (Spectrometer for Ice Nuclei, SPIN) and PINE (Portable Ice Nucleation Experiment), as well as filter collection for offline freezing analysis FRIDGE (FRankfurt Ice Nuclei Deposition FreezinG Experiment), INDA (Ice Nucleation Droplet Array), INSEKT (Ice Nucleation Spectrometer of the Karlsruhe Institute of Technology), IS (Ice Spectrometer), LINA (Leipzig Ice Nucleation Array), LINDA (LED based Ice Nucleation Detection Apparatus), and the UNAM-MOUDI-DFT (Universidad Nacional Autónoma de México Micro-Orifice Uniform Deposit Impactor–Droplet Freezing Technique); filters were collected and compared for consecutive 8 hours; online INP measurements are compared within a time span of 10 minutes; PINE partly joined the offline intercomparison, measuring at a constant temperature during the 8 hours; online instruments measured partly behind the Portable Fine Particle Concentrator (PFPC; Gute et al., 2019).

## 2.2 Online measurement techniques

Three different online INP instruments were operated behind the WAI in parallel for several hours per day. INP concentrations were determined for single particles activating in a temperature range between ~ -20 °C and -30 °C, in the condensation/immersion freezing mode (via controlling processing relative humidity). All INP concentrations are referenced to standard liters sampled. For the intercomparison of these instruments, INP

concentrations are only considered when measured within ± 1 °C and within ± 10 minutes. This aims to reduce any potential impact of a change in the sampled INP population at presumably nearly identical sampling conditions. Based on the PINE data collected during this campaign, an average increase of 1.7 in INP concentration was observed for an increase in nucleation temperature by 1 °C. This factor of ~1.7 is below the chosen factor of 2 to determine a good agreement between the online instruments. We acknowledge that a ± 1 °C range can lead to variations in detected INP concentrations, however, a more restrictive approach would further limit the amount of comparable data points.

During specific periods, online INP measurements were conducted downstream of the Portable Fine Particle Concentrator (PFPC; Gute et al., 2019), which is optimized for concentrating aerosol particles > 0.1 µm. The PFPC was deployed at a separate inlet and used an impactor with a 50% size cut at 2.5 µm. The inlet and outlet flows of the PFPC were kept at the same values as described by Gute et al. (2019), i.e., 250 LPM and 10 LPM, respectively. Aerosol particles are concentrated with factors of ~ 20 for particles > 0.5 µm, and with lower values for smaller particles. Since the INPs can be of sizes below and above 0.5 µm, INPs can be concentrated with variable factors (INP concentration factor). For the intercomparison between the online INP instruments, the same INP concentration factors were applied for simultaneous measurements. This did not have an impact on the instruments' comparability, given that the instruments did not use additional impactors smaller than the PFPC's impactor with a size cut of 2.5 µm. The INP concentration factor used for the online intercomparison is thereby a campaign average of 11.4 and has a standard deviation of 1.7. This INP concentration factor was inferred by consecutive measurements with the concentrator turned on and off sequentially, using CSU-CFDC, which performed such measurements most frequently. The average INP concentration factor derived with PINE was similar (campaign average 10.9) but with a higher standard deviation (5.8), which might arise from the fact that PINE does not use an impactor when not sampling at the concentrator, such that larger particles, that are ice-active, can enter the instrument and contribute to more variation of the measured INP concentrations. For the comparison to the filter-based offline INP concentrations, a daily average INP concentration factor from CSU-CFDC was used to convert concentrated to ambient INP concentrations when sampling from the PFPC. This daily average INP concentration factor ranged from values of 8.5 to 16.5, reflecting variability in the sizes of INPs present at different times.

The instrument specifications are summarized in Tab.1 1 and are explained in more detail in the following.

Table 1: Specifications of the online instruments.

| Name | CSU-CFDC | SPIN | PINE |
|---|---|---|---|
| inlet | WAI / PFPC* | WAI / PFPC* | WAI / PFPC* |
| impactor | two impactors with 2.5 µm size-cut | one impactor with 2.5 µm size-cut | no impactor; size-cut 4 µm (Möhler et al., 2021) |
| temperature and RH$_{water}$ uncertainty | ± 0.5°C and 2.4% | ± 0.5°C and 2.5% | ± 1 °C |
| residence time | 5 s | 10 s | < 33 s |
| supersaturation | 106.5% RH$_{water}$ | 102.8% RH$_{water}$ | > 100% RH$_{water}$ |
| ice threshold | 4 µm | 5 µm | automated |

\* online instrument sampled always at the same inlet

### 2.2.1    The Colorado State University Continuous Flow Diffusion Chamber (CSU-CFDC)

The CSU-CFDC is the longest-existing instrument for online detection of ambient INPs, with a legacy of versions for ground and aircraft-based measurements starting from the late 1980s (Rogers, 1988; Rogers et al., 2001; DeMott et al., 2018). Its working principle is based on the establishment of supersaturated water and ice conditions in flowing air between two ice-coated walls of cylindrical shape in a vertical orientation. Those walls are held at different temperatures, and while the air temperature in the central lamina region is a linear function between these

temperatures, the water vapour pressure is a non-linear function of temperature, resulting in a supersaturated region with respect to ice and water between the walls. The aerosol lamina is surrounded by particle-free sheath air through this region where particles can activate into water droplets and ice crystals. While cloud droplets are evaporated downstream using an evaporation section, the remaining ice crystals are detected by their larger size using an OPC (Climet CI-3100). The size threshold to determine ice crystals was thereby 4 μm. The CSU-CFDC

uses a pair of single-jet impactors upstream of the chamber, for this study with inserts defining 50% aerodynamic size-cuts at 2.5 μm, such that effectively only aerosol particles smaller than this size enter the system. This allows ice crystals to be differentiated from the larger ambient aerosol. The measurement uncertainties at -30 °C with regard to temperature and relative humidity with respect to water ($RH_{water}$) are stated as $\pm$ 0.5 °C and 2.4%, respectively (DeMott et al., 2015). The residence times of aerosols in the supersaturated region are 5 s for the flow

rate used (1.5 LPM; liter per minute). For this study, water supersaturation was controlled to be sufficiently high to promote comparison to the results of immersion freezing methods (DeMott et al., 2017). The mean and median supersaturations employed for this study were both equal to 6.5% (i.e., 106.5% $RH_{water}$), with a standard deviation of 1.4%. At this value, it is likely that maximum INP concentrations are not captured, although underestimations would be expected to be less than the factor of 3 noted for mineral dusts in comparing data collected at 105%

versus 109% in DeMott et al. (2015). The 1-Hz data were accumulated and averaged over a time period of 1 minute for this study. CSU-CFDC is typically operated for ~ 4 hours before refreshing the ice surfaces on the walls. Operation times in excess of 4 hours can result in an increase in background ice counts (due to frost) in the chamber and thereby degrade the signal-to-noise ratio. CFDC background corrections are needed to account for INP signal contamination that may come in the form of frost crystals flaking from the ice walls (Rogers et al. 2001).

Infrequent, high-concentration bursts may occur, typically in the time just following wall icing or after a number of hours of operation. These are accounted for with a data pre-screening method to search for outliers in ice crystal arrival rates at the optical particle counter (Moore, 2020). The more common intermittent, low-concentration frost events are corrected by comparing ambient measurements with measurements of HEPA-filtered air. For PICNIC, these filter periods were 5 minutes long, bookending each 10-minute ambient air sample period. The correction

for intermittent frost events has recently been modified to improve the estimates of statistical significance and confidence intervals over previous techniques, following Krishnamoorthy and Lee (2013). The background INP counts from filter periods that bracket each ambient measurement are combined into a single Poisson distribution with a characteristic rate parameter. The difference between the ice crystal arrival rates during the ambient measurement and the combined filter period is used to calculate the background-corrected INP concentrations

(Moore, 2020). Statistical significance and confidence intervals for each ambient measurement are determined using the moment-based Z-statistic defined in Krishnamoorthy and Lee (2013).

### 2.2.2    The Spectrometer for Ice Nuclei (SPIN)

The SPIN is a commercially-available CFDC-style instrument developed by Droplet Measurement Technologies (Garimella et al., 2016). It is based on the design of the laboratory instrument ZINC (Zurich Ice Nucleation Chamber; Stetzer et al., 2008) and its mobile version PINC (Portable Ice Nucleation Chamber; Chou et al., 2011). Briefly, two parallel flat plates are separated by 1 cm and each coated with 1 mm of ice prior to experiments. A temperature gradient between the two plates establishes a supersaturation with respect to ice and potentially liquid water. The supersaturation employed for this study was $2.8 \pm 1.9\%$ ($102.8\%$ $RH_{water} \pm 2.5\%$), with an uncertainty in temperature of $\pm 0.5$ °C. Aerosols are fed into the chamber at a sampling rate of 1 LPM and constrained to a lamina center-line with 9 LPM of sheath air. The residence time of the particles in the chamber is 10 seconds. An impactor with a 50% size cut at 2.5 µm (BGI Inc., SCC1.062 Triplex) was installed before the SPIN inlet. Activated INPs are detected using a light-depolarization OPC (Garimella et al. 2016; Droplet Measurement Technologies). Due to the sigmoidal shape of the impactor's size cut, OPC counts larger than 5 µm in diameter were considered as activated INPs. Although SPIN is operated at a lower supersaturation as compared to the CSU-CFDC, the ice crystals have a longer residence time (10 seconds) such that they grow to sizes larger than 5 µm.

Aerosol spreading due to turbulence at the inlet can cause some sampled aerosol to spread outside of the aerosol lamina, where they are exposed to a lower relative humidity. This phenomenon is ordinarily accounted for by applying measurable correction factors to the CSU-CFDC and SPIN data (DeMott et al. 2015; Garimella et al. 2017; Wolf et al. 2018). The degree of aerosol lamina spreading, and therefore the correction factor applied to observed INP concentrations, depends on several variables such as inlet pressure, chamber temperature, and degree of supersaturation. The correction factor for SPIN has been quantified to vary from approximately 1.5 to 9.5 for immersion-freezing conditions (Garimella et al. 2017). As the degree of aerosol lamina spreading was not quantified in this study, no correction factor was applied.

The uncertainty in INP concentration for SPIN represents the standard deviation during a 10-minute sampling period. SPIN's limit of detection is dependent on background ice concentrations resulting from ice shed from the walls. Backgrounds were measured for 5 minutes on both sides of a 10-minute sampling period. Average backgrounds before and after a sampling period were subtracted from the average measured INP concentration. Only data from when backgrounds were less than half of measured INP concentrations are reported. The campaign-averaged background concentration was $\sim 3$ $L^{-1}$. The limit of detection of SPIN sampling at the concentration is lower ($\sim 0.6$ INP $L^{-1}$) as compared to not sampling at the concentrator ($\sim 6$ INP $L^{-1}$), as more sampled air is analyzed, while the ice background counts remain the same. SPIN can typically be operated for four to six hours before backgrounds are too high to prevent measurement of ambient INP concentrations. Besides the results from SPIN presented in this manuscript, focusing on mixed-phase cloud conditions, SPIN also measured cirrus-relevant INP concentrations, which is discussed elsewhere (Wolf et al., 2020).

### 2.2.3    The Portable Ice Nucleation Experiment (PINE)

PINE is a new type of mobile instrument to measure INPs (Möhler et al., 2021). It is based on the AIDA (Aerosol Interaction and Dynamics in the Atmosphere) chamber and mimics cloud formation upon air mass lifting by expansion. The instrument is fully automated and can be operated continuously. During the PICNIC campaign, the PINE version PINE-1A was deployed. This version consists of a 7-liter cylindrical chamber, which is cooled by an external ethanol cooling chiller (Lauda RP 855; Lauda-Königshofen, Germany). PINE operates in a cycled

mode of flush, expansion, and refill. During the so-called flush mode, aerosol particles are guided through the chamber at a flow rate of 2 LPM for 5 minutes. Before entering the chamber, the sampled air is dried to a frost point temperature of below ~ -13 °C, which avoids accumulation of ice on the chamber wall. An OPC (welas-2500, Palas GmbH, Karlsruhe, Germany) attached to the outlet of PINE counts larger unactivated aerosol particles. The flush mode is followed by the expansion mode when a valve upstream of the chamber is closed while the volumetric flow out of the chamber is set to a constant value of 3 LPM. Please note that the inlet flow rate during the expansion is maintained by a bypass flow which is the same as the flush flow rate, such that no change in the sampling flow at the WAI occurs. A total pressure reduction of ~300 mbar is thereby induced over a time of ~50 s. During this expansion, the air temperature in the chamber is decreased by expansion cooling. As the wall and air temperatures are below the frost point temperature, the chamber is ice-saturated at the start of the expansion, and achieves supersaturation with respect to ice and water during the course of the expansion, such that cloud droplets (upon cloud condensation nuclei) and ice crystals (upon INPs) can form. The temperature during one expansion typically decreases by 6 °C. The coldest temperature is thereby used as the nucleation temperature for each experiment, as it is assumed that the coldest temperature dominates the INP number concentration. After completing an expansion, the chamber is set to the refill mode where the chamber is refilled with filtered sample air to reach ambient pressure conditions. Then another cycle of flush, expansion, and refill is started.

During the expansion, the ice crystals are detected by their comparably large optical size in the OPC, which makes a distinction with cloud droplets possible. As the OPC has a sideward scattering geometry, aspherical ice crystals are detected with a higher scattering intensity than spherical cloud droplets of the same volume and refractive index. No ice-background correction is needed for the INP measurements since no ice crystals form from frost forming on the walls, which is confirmed by regular background experiments when the sampled air is passed over a filter to remove all particles before entering the chamber for several consecutive expansions.

In the PINE instrument, the residence time of aerosol particles at supersaturated conditions or in supercooled droplets is more variable as compared to CFDC instruments. The time during which cloud droplets are present during an expansion is 33 seconds. However, it should be noted that this is an upper limit for the residence time, as ice crystals formed by INPs are detected during the whole expansion period and each INP has its own trajectory within the cloud chamber. In the presented study, the INP concentrations are averaged over two consecutive experiments (two cycles of flush, expansion, and refill) to increase the detection limit for INPs. During the course of one expansion, about 2 litres of air are continuously taken out of the chamber and analyzed for forming ice crystals. The welas-2500 OPC has an optical detection volume of 10%, thus having a limit of detection of 2.5 INP per liter for two consecutive experiments. The uncertainty for the INP concentration is 20%, which is an upper estimate from the uncertainties of the determination of the optical detection volume. The uncertainty in temperature is ± 1 °C (see Möhler et al. (2021) for further details about the specifications of PINE).

The majority of aerosol particles with an aerodynamic diameter of < 2 μm are sampled with PINE (80%), which decreases to < 50% for particles with an aerodynamic diameter of > 4 μm. No impactors were used with the PINE instruments. However, when sampling at the PFPC, which is operated with an impactor with a 50% size cut at 2.5 μm, the sampled particle size was limited to this size.

In order to compare the PINE measurements to the offline methods for a perfect time overlap, PINE joined the offline intercomparison times for some night-time measurements and measured at a constant temperature during the 8 hours.

## 2.3    Offline measurement techniques

For offline INP analysis, aerosol particles were collected simultaneously with the different sampling setups during 8-hour intervals. All INP concentrations are given with reference to standard liters sampled. Here, we present results from day- and night-time sampling periods (10 am to 6 pm and 10 pm to 6 am, respectively) from the 7th to the 20th of October 2018 (Table 1). Only during the 18th to 19th of October, the sampling time was increased to 24 hours. The particles were collected on filters, either behind the WAI (no additional impactor used) inside the laboratory or directly on the rooftop (Fig. 1). After collection, the samples were transported frozen or refrigerated to the respective laboratories, and particles were resuspended from the filters to analyze their ice nucleation activity in the immersion freezing mode. The comparison of the INP freezing spectra derived with the different methods is done at 1 °C intervals. A total of seven offline methods were deployed during PICNIC, which are described in the following sections, and their specifications regarding filter collection and freezing analysis are summarized in Table 1.

The cumulative INP concentration calculation as a function of the nucleation temperature $c_{INP}(T)$ for all offline techniques follow the well-established Vali (1971) equation:

$$c_{INP}(T) = \frac{V_{sus}}{V_{air}} \frac{1}{V_{drop}} \left( ln\left(\frac{N_{all}}{N_{l(T)}}\right) - ln\left(\frac{N_{all,BG}}{N_{l,BG}(T)}\right) \right) \tag{1}$$

Where $V_{drop}$ is the droplet volume, $N_l$ is the number of liquid and thus unfrozen droplets, while $N_{all}$ is the number of the total droplets containing the aerosol suspension. The calculation thereby considers the volume of water used to extract the sample (suspension; $V_{sus}$) and the volume of air sampled $V_{air}$ (considering the filter collection time and the applied flow rate). The number of total droplets from background measurements ($N_{all,BG}$) and the number of liquid droplets from background measurements $N_{l,BG}(T)$ are inferred from the freezing curves of field blank filters, which were handled the same way as the sample filters except that no air flow was guided over the blank filter. The INP errors are indicated by using two-tailed, 95 % confidence intervals for binomial sampling based on Agresti and Coull (1998).

### 2.3.1    FRankfurt Ice Nuclei Deposition FreezinG Experiment (FRIDGE)

For the FRIDGE measurements, aerosol particles were collected in the laboratory from the WAI inlet. Aerosol was collected by using a custom-built semi-automated multi-filter sampling device. The unit consists of 8 individual filter holders, the 45.7 cm housing, valves, a pump, and electronics. The sampling time of each filter can be programmed separately. The flow rate through the filters was determined to be 4.8 ± 0.4 Std LPM on average. This is more than 50% lower than the flow rate that was originally targeted due to a miscalibration and a leakage in the system. Accordingly, the flow rate needed to be corrected to the above-mentioned value and carries a rather high uncertainty. Aerosol particles were collected onto 47 mm hydrophobic PTFE Fluoropore Membrane

Table 2: Specifications of the offline freezing methods.

| | Name | FRIDGE | INSEKT | INDA* | IS | LINA* | LINDA | UNAM-MOUDI-DFT |
|---|---|---|---|---|---|---|---|---|
| **filter collection** | location | WAI | WAI | WAI | rooftop | see INDA | rooftop | WAI |
| | time interval | 8 hours (night), 4 hours (day) | 8 hours | 8 hours | 8 hours | *same as INDA* | 8 hours | 8 hours |
| | substrate | 47 mm PTFE fluoropore membrane filter, 220 nm pore size | 47 mm polycarbonate filters, 200 nm pore size | 47 mm polycarbonate filters, 200 nm and 800 nm pore size; 47 mm quartz fiber filters | 47 mm polycarbonate filters, 200 nm pore size | *same as INDA* | 15 cm quarz fiber filters | hydrophobic glass coverslips |
| | filter holder | custom-built semi-automated multi-filter sampling device | standard | standard, HERA | open-faced sterile Nalgene sampling heads | *same as INDA* | high volume sampler | MOUDI cascade impactor |
| | flow | 4.8 LPM | 11 LPM | standard: 12 - 37 LPM HERA: 15-41 LPM | 13.5 LPM | *same as INDA* | 500 LPM | 30 LPM |
| | limit of detection (L$^{-1}$) | 4.3E-04 (8 hours) | 1.90E-04 | standard: 1.7E-04 - 5.6E-05 HERA: 1.4E-04 - 5.1E-05 | 1.5E-04 | *same as INDA* | 4.2E-06 | 6.9E-05 |
| | filter storage | partly unfrozen | frozen | frozen | frozen | *same as INDA* | frozen | refridgerated |
| **analysis** | liquid volumes | 2.5 µl droplets | 50 µl suspension | 50 µl suspension | 50 µl suspension | 1 µl droplets | 200 µl suspension | 100 µm droplets |
| | cooling rate | 1 °C min$^{-1}$ | 0.3 °C min$^{-1}$ | 1 °C min$^{-1}$ | 0.3 °C min$^{-1}$ | 1 °C min$^{-1}$ | 0.3 °C min$^{-1}$ | 10°C min$^{-1}$ |

\* INDA and LINA use the same collected filter

370

Filter of 220 nm pore size (Merck Millipore). Filters were not pre-cleaned in any way. It was decided to limit the sampling time for FRIDGE to 4 hours during daytime (10 am – 2 pm, i.e., termination in the middle of the total sampling time of other instruments), as we initially expected higher INP concentrations compared to night-time sampling, and to better capture potential variability in INP concentrations. The night-time sample was the same as for the other groups (8 hours). Moreover, on October 18[th], the sampling time was not increased to 24 hours, as for other methods. Filters were stored frozen at -18 °C after collection at the site. The samples were not actively cooled during transport, however, given the relatively short travel time of ~8 hours to the laboratory in Frankfurt, we do not consider that this impacts the results, but cannot be excluded for certain (Beall et al., 2020). After transport, they were stored in a refrigerator at 4 – 7 °C until freezing measurements were performed. The analysis was performed using the droplet freezing mode of FRIDGE (Hiranuma et al., 2015). Before starting a measurement, a filter containing the sampled aerosol was placed in a sterile Eppendorf tube, which was filled with 5 mL of ultrapure water (Rotipuran ultra, Carl Roth). Particles were then extracted into the ultrapure water by repeated steady shaking for several minutes, without dilutions. Using an Eppendorf Reference 2 pipette, a total of about 200 (184 – 231) 2.5 µL droplets were manually pipetted onto a 45 mm silanized (Dichlordimethylsilan) silicon wafer substrate placed on a cold stage inside of a 500 cm$^3$ measurement cell. About 65 droplets of 2.5 µL fit onto the substrate at a time, therefore three individual runs per sample were performed to improve the freezing statistics. Before and after each measurement run, the substrate was thoroughly cleaned with pure non-denatured ethanol (Rotipuran, >99.8 %, Carl Roth). During the experiment, the measurement cell was constantly flushed with dry synthetic air at 1 LPM to prevent condensation and riming. The temperature was decreased at a constant rate of 1 °C min$^{-1}$ until every droplet was frozen using a PID-controlled Peltier element. An ethanol cryostat cooling system supported the Peltier by dissipating the heat. The surface temperature was measured with a Pt100 sensor, which has an accuracy of ±0.2 °C. A camera saved measurement images every 10 s and the change in brightness was detected when droplets were freezing. A detailed description of the FRIDGE immersion freezing method can be found in Schrod et al. (2020).

395

### 2.3.2 Ice Nucleation Droplet Array (INDA)

For analysis with INDA and also the below discussed LINA (Leipzig Ice Nucleation Array, see section 2.3.5), three different types of filters and two different samplers were deployed, with both samplers operating in parallel. All filters were taken at the WAI. Quartz fiber filters (Munktell, MK 360; 47 mm diameter) were used for sampling, as well as polycarbonate filters (Nuclepore, Whatman, 47 mm diameter with pore sizes of 200 or 800 nm). One sampler was a simple standard filter holder. The sampling flow was deliberately set to different values for different sampling periods, varying between 12 and 37 LPM, resulting in total collected air volumes between 6 and 18 m$^3$. The other sampler was HERA (High Volume Aerosol Sampler, Hartmann et al., 2020; Grawe et al., 2023), which was developed for airborne sampling and enables the subsequent sampling of six filters. For HERA, the sampling flow was varied between 15 and 41 LPM, resulting in collected air volumes between 7 and 20 m$^3$. All samples and blank filters were stored in separate Petri-dishes right after sampling and stored and shipped frozen until the analysis was done at the laboratory in Leipzig.

INDA is based on a measurement technique that was introduced by Conen et al. (2012) and modified as suggested by Hill et al. (2014). A suspension is obtained by washing particles off a polycarbonate filter. For this, the filters are put in 3 mL of ultra-pure water, followed by shaking for 15 min in a flask shaker. Subsequently, typically 0.1 mL of the suspension is used for a LINA experiment (Sec. 2.3.5). Then 3.1 mL of ultra-pure water is added, and 50 μL droplets of this suspension are placed into 96 wells of a PCR tray. For the quartz filter samples, each well is filled with 50 μL of ultra-pure water together with a 1 mm diameter filter punch from the quartz fiber filter. The PCR tray is then immersed in a temperature-controlled cooling bath of a thermostat and is illuminated from below. During cooling, typically done at 1 °C min$^{-1}$, a picture is taken every 6 s from above. Changes in the color of wells occur during freezing and are automatically detected. More information can be found in Gong et al. (2020) for the INP analysis of quartz fiber filters and in Hartmann et al. (2020) for polycarbonate filters.

### 2.3.3 The Colorado State University Ice Spectrometer (IS)

The Colorado State University (CSU) Ice Spectrometer (IS) analyses arrays of liquid suspensions from filter samples to quantify immersion freezing INP concentrations (e.g., DeMott et al., 2018). Aerosol filter samples were collected on the roof of the laboratory using precleaned (5% H$_2$O$_2$ followed by two 100 nm - filtered deionized (DI) water rinses), 200 nm pore diameter, 47 mm diameter Nuclepore polycarbonate filter membranes (Whatman, GE Healthcare) held in open-faced sterile Nalgene sampling heads. Mass flow rates (at 101.3 kPa and 0 °C) were recorded at the start and stop of the sample period to calculate the total volume filtered. The average sample volume collected was 6 m$^3$. Filter samples were immediately placed into sterile Petri dishes (Pall) and stored and transported frozen until analysis of INPs in Fort Collins, Colorado.

For analysis, 10 mL of 0.1 μm-filtered (Whatman Puradisc, PTFE membrane) DI water was added to a pre-rinsed polypropylene 50 mL tube (Corning) and shaken in a Roto-Torque rotator (Cole-Parmer) for 20 minutes to create a suspension. For each sample, serial 20-fold dilutions were made to 8000-fold. Next, thirty-two 50 μL aliquots of each sample, corresponding dilutions, and a 0.1 μm-filtered DI water blank were dispensed into 96-well PCR trays (OPTIMUM® ULTRA Brand from Life Science Products) in a laminar flow hood. The trays were then placed into aluminum blocks in the IS and cooled at a rate of ~ 0.33 °C min$^{-1}$. Freezing was detected by a CCD camera and the corresponding temperature was recorded with a LabVIEW interface. Frozen fraction results were corrected for the number of INPs in the DI water blank, resulting in the lowest freezing temperature achievable (generally

between -27 and -30 °C). Temperature uncertainty is estimated at < ±0.5 °C. The proportion of frozen wells was converted to a number of INPs mL$^{-1}$ of suspension using Eq. 13 in Vali (1971) and subsequently scaled to the number of INPs per filter. The average number of INPs on three field blanks (cleaned, handled, transported, and processed in the same way with the exception of air flow) was subtracted from all samples before conversion to INPs per L of air considering the volume collected. Two-tailed, 95 % confidence intervals for binomial sampling

were calculated based on Agresti and Coull (1998). Some samples of IS were investigated for the size of INPs by filtering the suspensions at 3 or 0.8 μm.

### 2.3.4     The Ice Nucleation Spectrometer of the Karlsruhe Institute of Technology (INSEKT)

The INSEKT is a rebuild of the IS freezing method (e.g., Schneider et al., 2021). During PICNIC, aerosol particles were collected in the laboratory via the WAI with a standard filter holder. The aerosol particles were sampled with

a flow rate of 11.3 (±0.2) Std LPM on 47 mm diameter Nuclepore filters (Whatman) with a pore size of 200 nm. The filters were pre-cleaned (10% H$_2$O$_2$ solution) and kept frozen after aerosol particle collection and during transport until analyzed in the laboratory in Karlsruhe. For INSEKT analysis, aerosol particles are washed-off the filter using 8 mL filtered nanopure water (100 nm pore diameter filter and 18MΩ deionized water), and shaken on a rotator for 20 minutes to ensure the release of all particles from the filter. The resulting suspension is then diluted

by factors of 1, 15, and 225, and volumes of 50 μL are placed in wells of a sterile PCR tray, alongside filtered nanopure water samples to determine its freezing behaviour for a background correction. The PCR tray is then placed in an aluminum block cooled with an ethanol cooling bath (LAUDA RP 890; Lauda-Königshofen, Germany). From a starting temperature of 0 °C, the wells are cooled down at a rate of 0.33 °C min$^{-1}$. Four Pt100 temperature sensors are placed inside the aluminum blocks for each PCR tray, measuring with an accuracy of ±0.1

°C and a deviation to the edges of the wells of ±0.1 °C, resulting in an uncertainty in temperature of ±0.2 °C. A camera detects brightness changes of the wells that correspond to their freezing.

Washing water of handling filter blanks that were taken prior to the 8[th] of October 2018 started to freeze at -7 °C, which was traced back to using non-powder-free gloves during the filter handling procedure at the Puy de Dôme, which was changed thereafter, demonstrating the need to work cleanly (Barry et al., 2021). Therefore, filters

handled with non-powder-free gloves had to be disregarded. Moreover, filters containing parts of insects, which were sampled due to a leak in the WAI mesh, were excluded from the analysis.

### 2.3.5     The Leipzig Ice Nucleation Array (LINA)

LINA is based on a method described by Budke and Koop (2015). The filters were sampled as described in Sec. 2.3.2, however, only polycarbonate filters were analyzed in LINA using washed suspensions. Of the resulting

suspensions from the filter washing water, 90 droplets with a volume of 1 μL are pipetted onto a hydrophobic glass plate, which is placed on a Peltier element. Each droplet is contained in a separate compartment which is covered by a second glass slide. Droplets are illuminated by a ring of light installed above, together with a camera. During the cooling process, typically done at 1 °C min$^{-1}$, a picture is taken every 6 s from above. Changes in the reflection of the light by the droplets related to freezing are automatically detected. A more detailed description can be found

in Gong et al. (2019).

### 2.3.6 The LED based Ice Nucleation Detection Apparatus (LINDA)

The LED-based Ice Nucleation Detection Apparatus (LINDA) is an immersion freezing detection device that allows automatic detection of freezing in closed tubes by light transmission and is described in detail by Stopelli et al. (2014). Quartz filters (15 cm diameter) were used for analysis with LINDA, taken with a high-volume sampler at the rooftop with a sample flow of 500 LPM. The filters were stored in the freezer at -20 °C until analysis in the laboratory of LaMP close to the Puy de Dôme.

For analysis, 4 circular samples (1.2 cm diameter) were extracted from each filter and were washed in a 25 mL solution of 0.9% NaCl for 20 minutes, then 200 µL of the resulting solution was introduced in each of the 52 tubes. The array of tubes is placed in a cooling bath, with a Pt100 temperature probe at each corner of the array. A camera placed above the array detects the freezing of the tubes through the variation of intensity of the transmitted light through the tubes. Errors bars were calculated from freezing events from background filters and the NaCl solution.

INP concentration measurements from LINDA were already presented by Bras et al. (2022) to investigate the seasonal variability. Here, we focus on the comparison to other INP concentration measurements.

### 2.3.7 The Universidad Nacional Autónoma de México-Micro-Orifice Uniform Deposit Impactor–Droplet Freezing Technique (UNAM-MOUDI-DFT)

Aerosol particle collection was carried out by an inertial cascade impactor (MOUDI 100R, MSP) which divides the particles according to their aerodynamic diameter in each of its 8 stages (cut sizes: 0.18, 0.32, 0.56, 1.0, 1.8, 3.2, 5.6, and 10.0 µm). For this study, particles impacted on stages 2 to 7 were used. Hydrophobic glass coverslips (Hampton Research) were used as substrates in each of the 8 stages. During PICNIC, the collection of particles was done in the laboratory via the WAI at a flow rate of 30 LPM. After particle collection, the samples were stored in 60 mm Petri dishes and refrigerated at ~4 °C for transport to the laboratory in Mexico City, where the analysis using the droplet freezing technique (DFT) was performed. We note that storing samples for a longer transportation time might impact the INP concentration (e.g., Beall et al., 2020). Although we did our best to keep the samples below 0 °C by transporting them in a freezer with ice packs, it is very likely that the samples may have experienced temperatures slightly above 0°C right before reaching their final destination.

The DFT, built at the Institute for Atmospheric Science and Climate Change at the UNAM (Córdoba et al., 2021), is based on the design by Mason et al. (2015) and determines the concentration of INPs as a function of temperature and aerodynamic particle size via immersion freezing. Each substrate is isolated in a temperature-controlled cell. Supersaturated conditions with respect to water are generated to trigger cloud droplet formation on the aerosol particles deposited on the substrate. The typical size of the droplets is around 100 µm and 30 to 40 droplets are formed in the study area (1.2 mm$^2$). The experiment is monitored in real-time with an optical microscope (Axiolab Zeiss, Germany) with a 5 × / 0.12 magnification objective coupled to a video camera (MC500-W, JVLAB). Droplets are subsequently cooled down from 0 to -40 °C at a cooling rate of 10 °C min$^{-1}$. The temperature at which each droplet freezes is determined when the temperatures from the cold cell (monitored with a resistance temperature detector RTD, ±0.1 °C uncertainty) and the videos are integrated. The INP concentration is derived from the following expression from Mason et al. (2015):

$$[INPs(T)] = -ln\left(\frac{N_u(T)}{N_0}\right) \cdot \left(\frac{A_{deposit}}{A_{DFT}V}\right) \cdot N_0 \cdot f_{ne} \cdot f_{nu,0.25-0.10mm} \cdot f_{nu,1mm} \tag{2}$$

where $[INPs(T)]$ is the INP concentration, $N_u(T)$ is the unfrozen droplets (L$^{-1}$) at a certain temperature T (°C), $N_0$ is the total number of droplets analyzed, $A_{deposit}$ is the total area where the aerosol was deposited on the MOUDI hydrophobic glass coverslips (cm$^2$), $A_{DFT}$ is the area analyzed by the DFT, $V$ is the volume of air sampled by the MOUDI (L), $f_{nu}$ is a correction factor (dimensionless) that takes into account changes in deposit inhomogeneity in a range between 0.25–0.10 mm in each of MOUDI sample, and $f_{ne}$ is a correction factor, that varies between 1.2 and 4.7, and that takes into account the uncertainty associated with the number of nucleation events in each experiment.

## 3 Results and discussion

### 3.1 Intercomparison of online instruments

INP concentrations as measured with CSU-CFDC, SPIN, and PINE were typically intercompared from the morning hours to the late afternoon, at ice nucleation temperatures ($T_{nucleation}$) from -20 to -30 °C. Measurements were performed either directly at the WAI, or downstream of the PFPC attached to the WAI when INP concentrations were calculated back to ambient conditions (see section 2.2). As an example, Fig. 2 shows a typical day of intercomparison, the 11$^{th}$ of October. In the morning hours, the instruments were set to the start conditions ($T_{nucleation}$ = -21 °C), which was changed consecutively for every few hours by 2 to 5 °C. As seen from this intercomparison day, the instruments measure similar INP concentrations at similar $T_{nucleation}$, with deviations within the same order of magnitude.

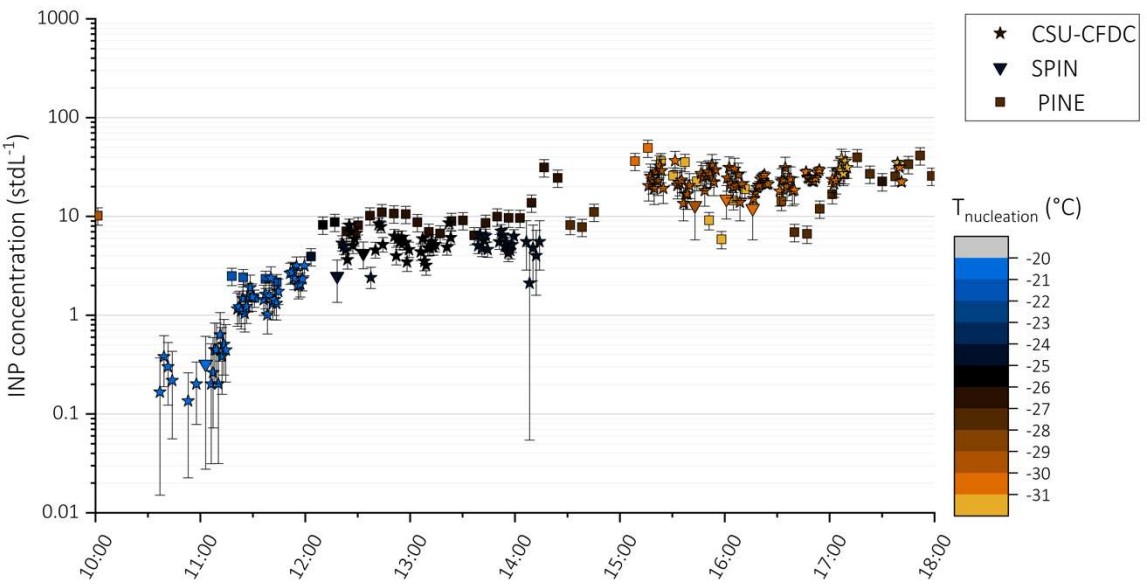

Figure 2: Time series of INP concentration above liquid water saturation as measured with CSU-CFDC (star), SPIN (triangle), and PINE (square) during the 11$^{th}$ of October 2018; the color scale represents $T_{nucleation}$; INP concentrations are measured with a time resolution of ~1 minute (CSU-CFDC) and ~10 minutes (PINE, SPIN).

To identify potential systematic deviations between the three instruments, the results from all intercomparison experiments are investigated using the CSU-CFDC as a reference instrument, given its long history of operation

and good characterization. CSU-CFDC has been used extensively in laboratory intercomparisons (e.g., DeMott et al., 2011; Hiranuma et al., 2015; DeMott et al., 2018) and in a large number of field measurement studies in surface- and aircraft-based campaigns (e.g., within the last five years, DeMott et al., 2018; McCluskey et al., 2018; Cornwell et al., 2019; Hiranuma et al., 2019, Kanji et al., 2019; Levin et al., 2019; Schill et al., 2020; Barry et al., 2021; Knopf et al., 2021; Twohy et al., 2021) over a period of more than 25 years. However, it should be noted that also the CSU-CFDC might not measure the total ambient INP concentration, due to aerosol lamina properties and size cuts, which will be discussed below in more detail and that can lead to an underestimation of the INP concentration. For the comparison with the SPIN and PINE, the CSU-CFDC data, which has the highest time resolution of 1 minute, were integrated on the time grid of the other instruments. Moreover, only measurements within ± 1 °C were considered. INP concentrations as measured with SPIN (Fig. 3a) and PINE (Fig. 3b) are compared against CSU-CFDC at a large dynamic range of INP concentrations (0.1 – 100 INP stdL$^{-1}$). This comparison reveals that SPIN observed lower INP concentrations, independently of $T_{nucleation}$. While only 35% of the data are within a factor of 2, 80% are still within factor 5 (Table 3a). It should be noted that only 20 data points could be compared here due to the mentioned temperature and time restraints. A possible explanation for this systematic deviation could be related to the aerosol lamina properties. Previous studies have found that the aerosol particles in at least some CFDCs are likely spreading beyond the lamina, such that not 100% of particles are in the lamina where they are exposed to the targeted supersaturation condition (DeMott et al., 2015; Garimella et al., 2017; Wolf et al., 2019). The issue of lamina spreading is likely variable and depends on the CFDC geometry, the flow conditions, and the temperature gradients between the walls, which is creating the supersaturation; ultimately this may be an issue with how the central lamina is introduced to the chamber, and how the thermal gradients and non-laminar flow at the location where the aerosols are entering the chamber impact their spreading. Aerosol spreading causes aerosol particles to experience lower supersaturations than the target supersaturation, resulting in either a non-activation into cloud droplets and ice crystals (immersion freezing mode) or an activation into ice

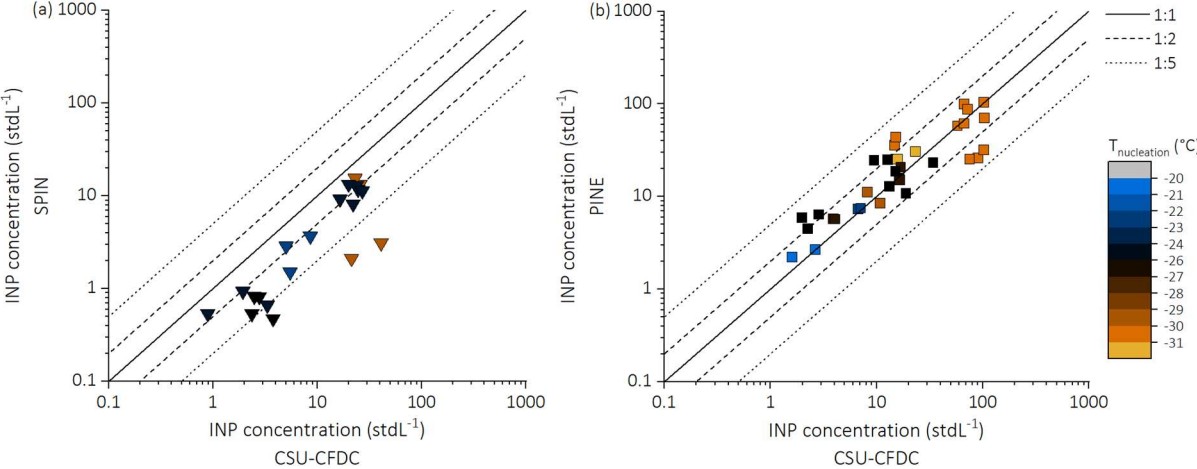

Figure 3: Comparison of INP concentrations measured with the SPIN (a), PINE (b) against CSU-CFDC; INP measurements are selected for cases that fall within ± 1 °C and overlapping sampling time; the measurements are corrected for the use of the aerosol concentrator when instruments sampled on it, by applying a correction factor of 11.4, which is the campaign average determined by CSU-CFDC.

crystals that are not growing to sizes within the residence time in the chamber to be detected by the OPC (above the ice threshold). SPIN was operated at a lower supersaturation ($2.8 \pm 1.9\%$) as compared to CSU-CFDC ($6.5 \pm 1.4\%$). Thus, it is expected that SPIN underestimates INP concentration, by up to a factor of 10 (Garimella et al., 2017; Wolf et al., 2020).

Moreover, SPIN also used a larger ice threshold in the OPC of 5 µm, against 4 µm from CSU-CFDC, which has been found to impact INP concentration measurements (Jones et al., 2011). Thus, it is possible that due to a larger ice threshold size, fewer particles in SPIN were encapsulated in the intended conditions, and were less likely to reach the critical size threshold. The impact of aerosol spreading was not quantified during the campaign, and data reported for the CSU-CFDC and SPIN instruments here remain original to account for this phenomenon. Moreover, no laboratory-derived calibration factors to account for a possible underestimation were applied, as the aim was to investigate such potential deviations amongst instruments using ambient aerosol particles. Please note that the residence time of SPIN is longer (10 seconds) as compared to CSU-CFDC (5 seconds), however, we believe that other factors such as the difference in supersaturation are more important here.

The comparison between CSU-CFDC and the expansion chambers PINE shows that the majority of the compared data fall within a factor of 2 (71%) and 5 (100%; Fig. 3b, Table 3b). As seen in Fig. 3b, no trend for under- or overcounting is observed for PINE relative to the CSU-CFDC. However, it should be noted that agreement between the measurements does not necessarily imply that both instruments can quantify the true ambient INP concentration. As stated before, the INP concentration using the CFDCs could be underestimated due to the incomplete activation of INPs in the aerosol lamina. The expansion chamber PINE could also systematically underestimate INP concentrations as it is possible that not all sampled aerosols are activating into cloud droplets, e.g., by being poor cloud condensation nuclei. More laboratory experiments will be performed in future studies to identify such a possible low bias. In addition, the residence time of particles in PINE is not as well quantified as in the CSU-CFDC, and might be longer (maximum 33 seconds), which might impact INP concentrations. It should also be pointed that, due to a temperature calibration performed after the PICNIC campaign, the PINE had fewer overlapping measurements with CSU-CFDC as initially targeted.

It should be noted that differences between the online instruments might arise from the difference in impactors. CSU-CFDC is operated with two single-jet 2.5 µm impactors, while SPIN is using only one, and PINE is operated without an impactor and thus has a 50% aerodynamic size-cut at 4 µm due to the loss of particles in its inlet.

Table 3: Comparison between the online methods (a; reference to CSU-CFDC) and offline methods (b, reference to INSEKT).

| (a) | method compared to CSU-CFDC | # compared data | Within a factor 2 (%) | Within a factor 5 (%) |
|---|---|---|---|---|
| | SPIN | 20 | 35 | 80 |
| | PINE | 34 | 71 | 100 |

| (b) | method compared to INSEKT | # compared data | Within a factor 2 (%) | Within a factor 5 (%) |
|---|---|---|---|---|

| | | | |
|---|---|---|---|
| FRIDGE | 259 | 46 | 88 |
| UNAM-MOUDI-DFT | 103 | 45 | 77 |
| LINA | 147 | 49 | 87 |
| INDA | 95 | 45 | 91 |
| IS | 300 | 27 | 65 |
| LINDA | 26 | 19 | 85 |

## 3.2    Intercomparison of offline methods

INP concentrations were determined based on 8-hour day- and night-time filter samples during the campaign, using seven different freezing methods. Due to the difference in sampled volume and thus detection limit (see table 2), the probability to detect very rare INPs at temperatures above ~ -10 °C varies amongst instruments. The time series of INP concentrations from those measurements are presented in Fig. 4 at key temperatures where many methods determined INP concentrations. Over temperatures ranging from -10 °C (Fig. 4a), -15 °C (Fig. 4b), and -20 °C (Fig. 4c), INP concentrations vary over three orders of magnitude, yet the measurements with a number of the different methods at single temperatures are most of the time within the error bars of each other. One clear systematic difference is that the rooftop INP concentrations (IS and LINDA) were systematically higher than those behind the WAI (all the other measurements), whereas the measurements taken from behind the inlet were generally within the quoted error bars. In order to get a more detailed picture of the results from the offline methods, the freezing spectra from each method for all day- and night-time samples are shown in Figs. 5, 6, and 7 (alongside the online data). The INP concentrations from the offline methods were determined between ~ -5 °C and -30 °C, and span a range from below 0.001 to above 100 INP stdL$^{-1}$. For most sampling intervals, the methods show good agreement, and the INP concentration and the shape of the freezing spectra are within a factor of 10. This is an indication of the general suitability of the different analysis procedures to determine INP concentrations (droplet freezing on cold stages, freezing of suspensions, using different cooling rates), and that the different filter holders (standard filter holders, FRIDGE custom-built semi-automated sampler, open-faced disposable Nalgene units, MOUDI sampler, HERA) and the filter materials (PTFE fluoropore membrane filters, quartz filters, hydrophobic glass coverslips, polycarbonate filters (200 and 800 nm pore diameters, see also section 3.2.2) can be used for INP collection. As mentioned, the IS and LINDA tend to measure higher INP concentrations, which appears to be associated with their filter sampling location on the rooftop, rather than from the WAI. Moreover, the INP concentration determined with the online instruments generally agrees to the offline freezing spectra (Figs. 5, 6, and 7) when sampling from the WAI, which will be discussed in more detail in section 3.3.

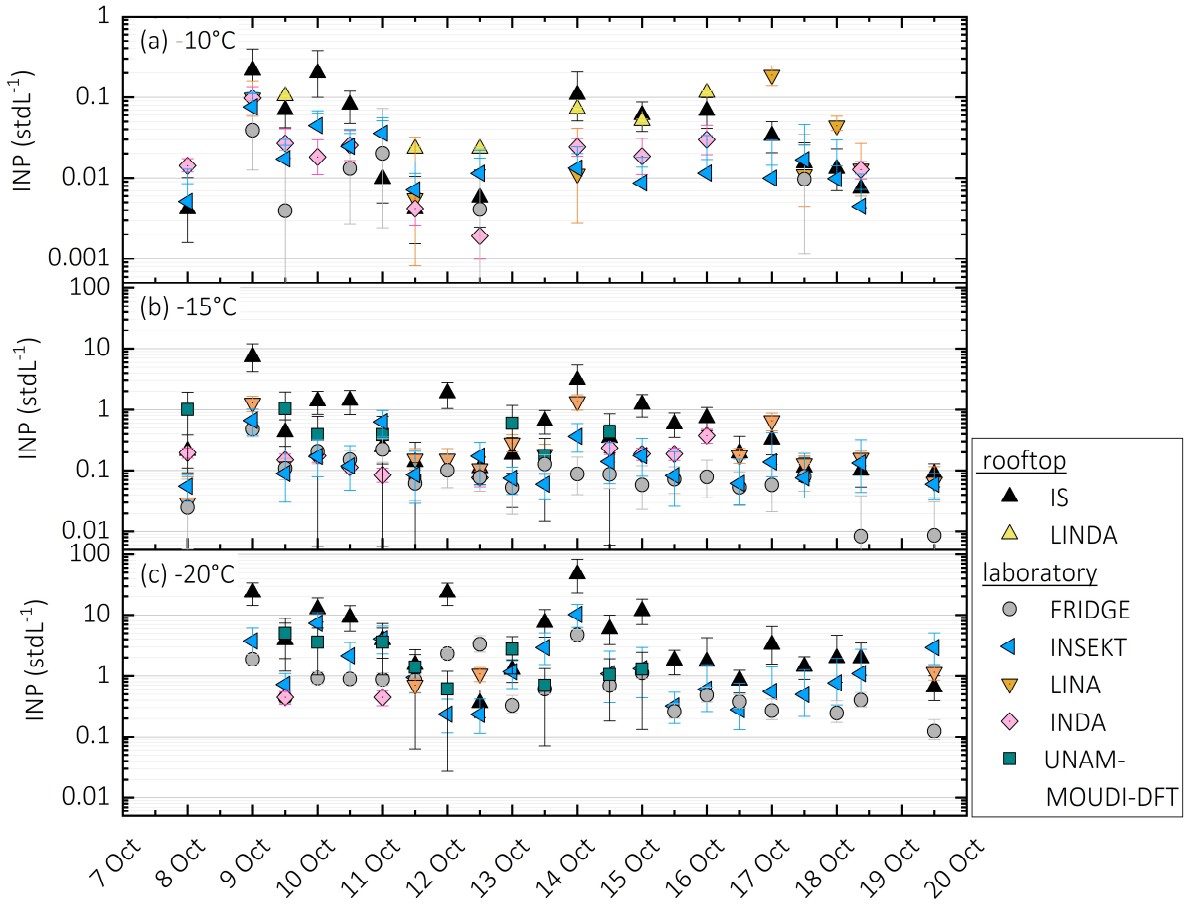

Figure 4: Time series of INP concentrations at -10 °C (a), -15 °C (b), and -20 °C (c) as measured with the offline techniques on the rooftop (IS, LINDA) and in the laboratory at the WAI (FRIDGE, INSEKT, LINA, INDA, and UNAM-MOUDI-DFT).

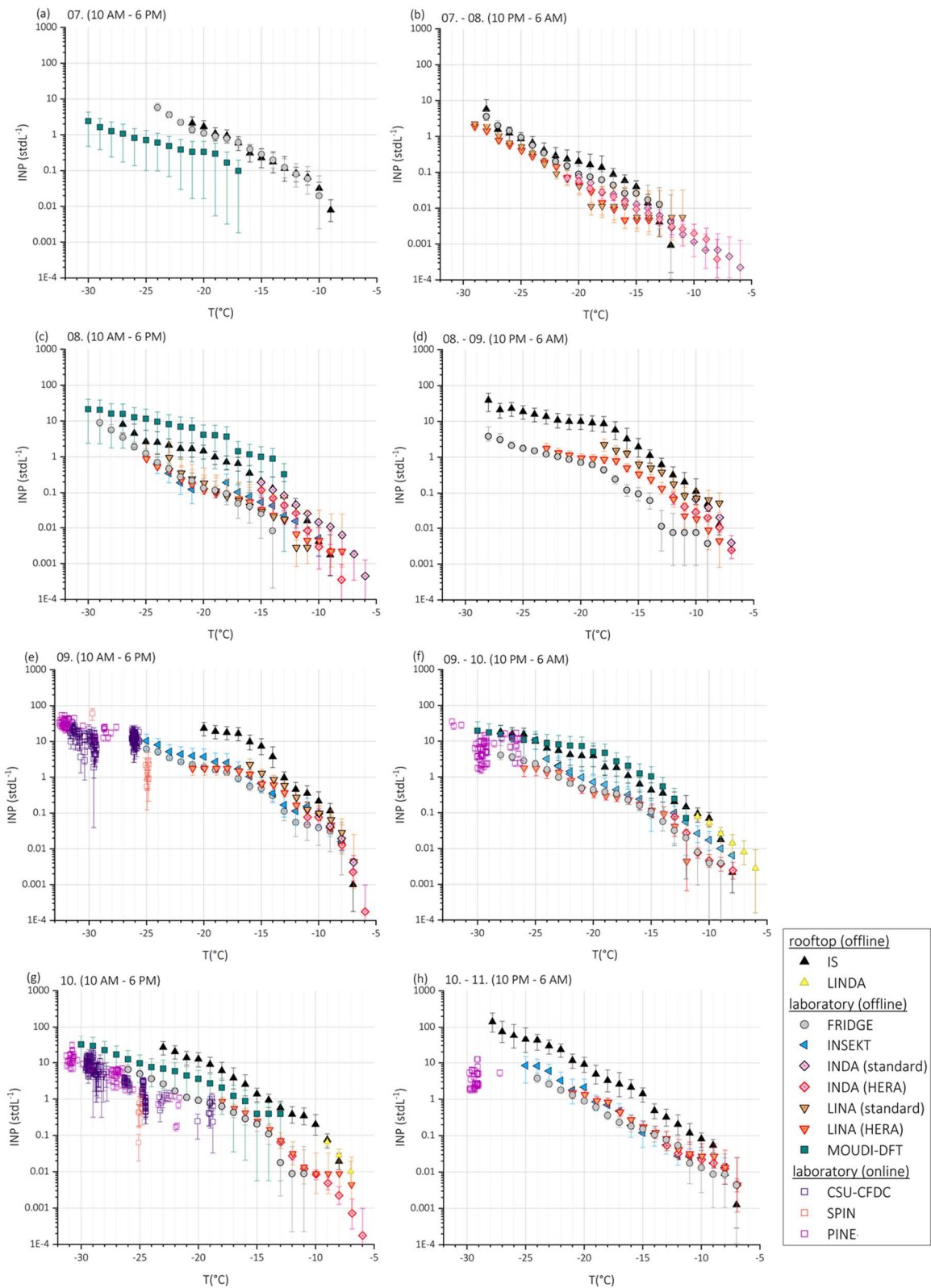

Figure 5: INP freezing spectra of the offline and online methods during the sampling time 7[th] to the 10[th] of October 2018; the filters for the offline INP analysis were taken during an 8-hour interval, except FRIDGE during the daytime samples (10 am – 2 pm); INP concentrations with the online instruments were determined within the same sampling period, but with a higher time resolution of minutes. Particles were collected on quartz filters for INDA and LINA using the standard filter holder (e - h).

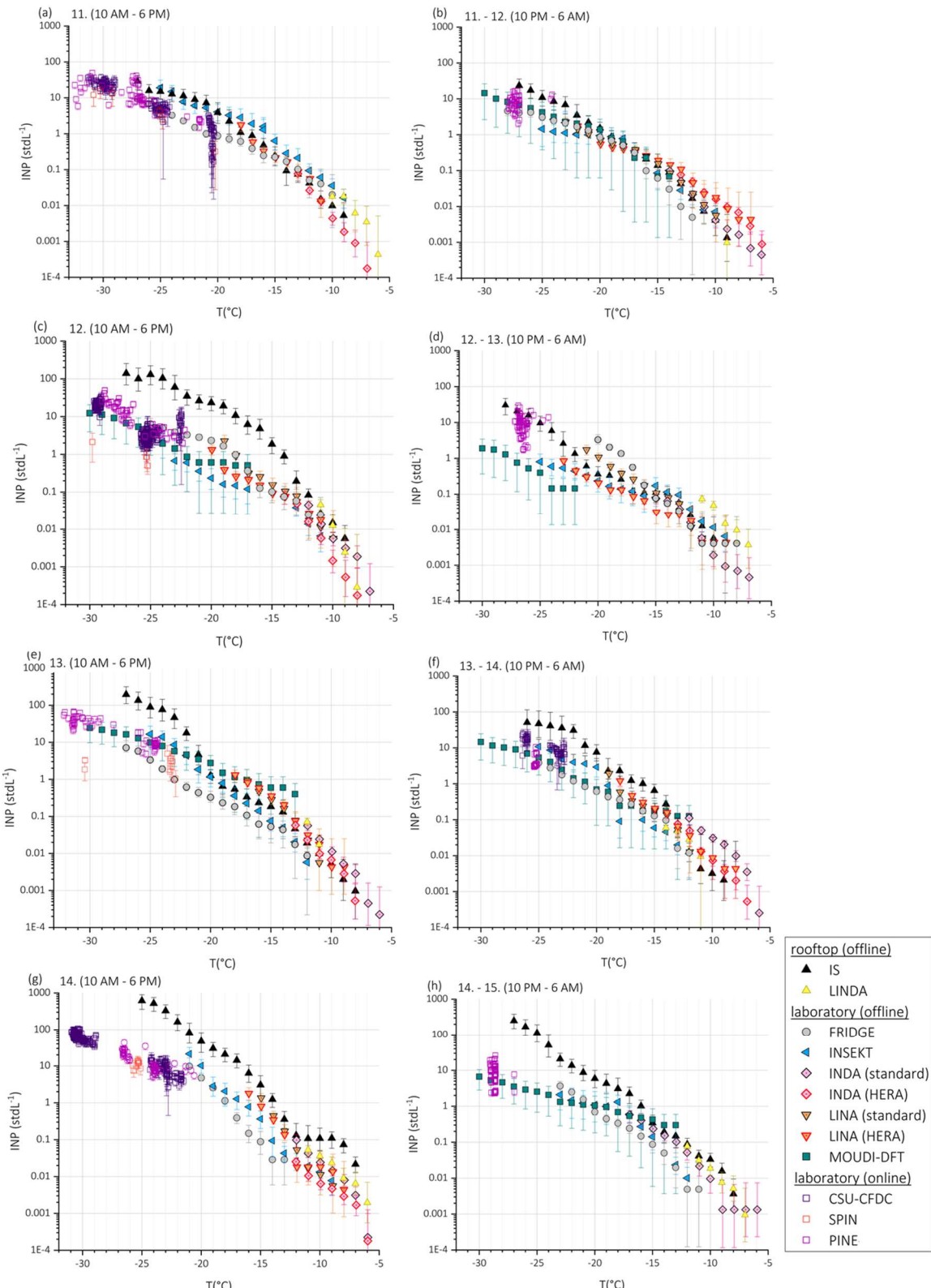

Figure 6: INP freezing spectra of the offline and online methods during the sampling time 11th to the 14th of October 2018, see description of Fig. 5. Particles were collected on quartz filters for INDA and LINA using the standard filter holder (a).

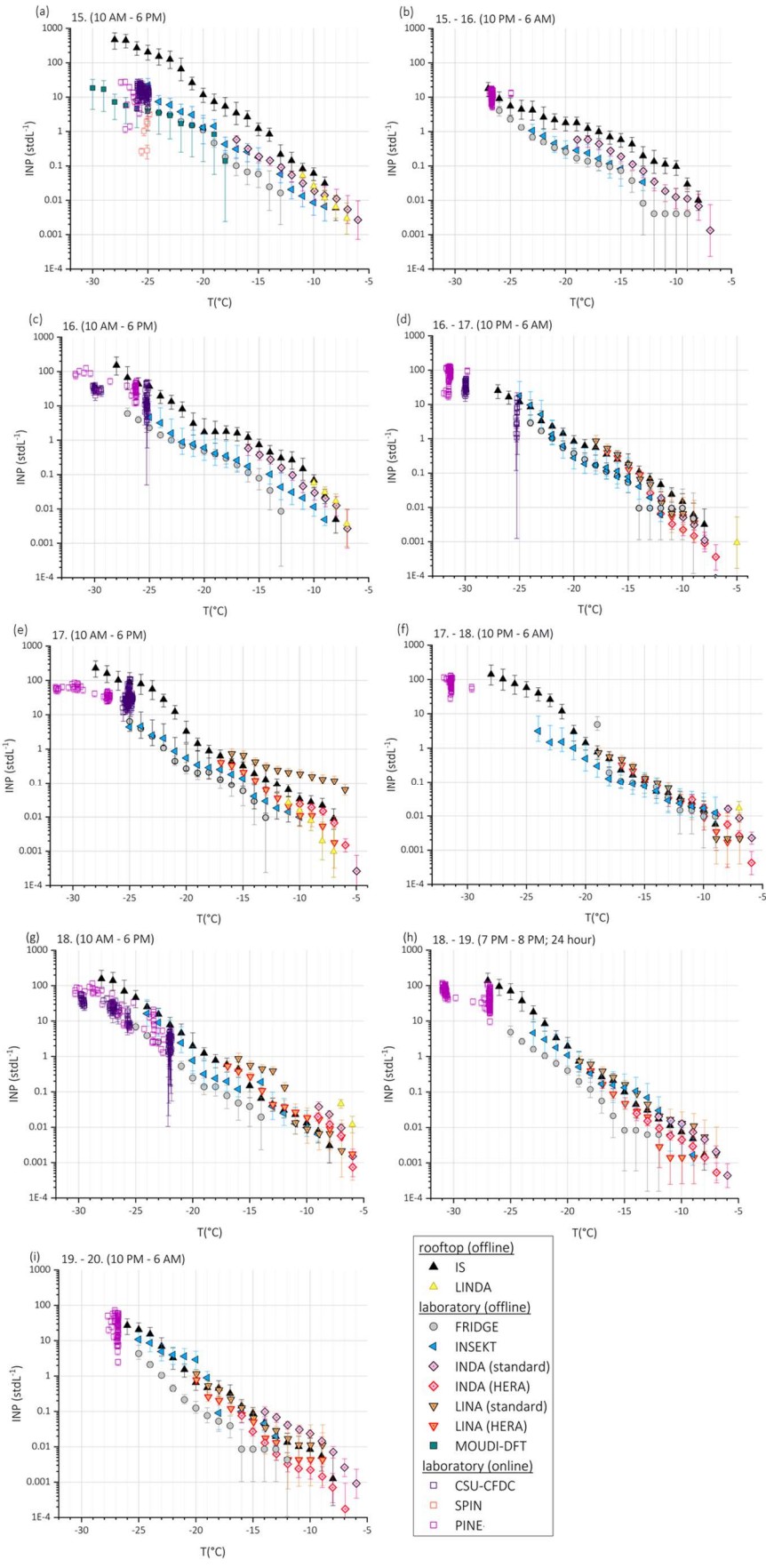

Figure 7: INP freezing spectra of the offline and online methods during the sampling time 15[th] to the 19[th] of October 2018, see description of Fig. 5. The standard filter holder and HERA for analysis with INDA and LINA were equipped with quartz fiber filters (a – c) and polycarbonate filters with a pore size of 800 nm (d - g).

In order to get better insight into the agreement during the whole campaign, we present the freezing spectra from each method compared against the INSEKT measurements as a reference (Fig. 8). This method was chosen since filter collection for INSEKT was performed in the laboratory at the WAI inlet, similar to that for most of the other methods, and since it covers a large temperature range (approximately from -8 to -25°C) of INP measurement. Figure 8 includes only data for INDA and LINA obtained from the standard filter holder, as no influence from the two different samplers (standard and HERA) was observed (see Figs. 5 – 7). Comparisons to INSEKT results on an instrument-by-instrument basis reveal that the methods sampling filters at the WAI on average agree with INSEKT for > 45% of the data within a factor of 2 and for > 77% within a factor of 5 (Table 2b). The FRIDGE method (Fig. 8a) has a slight tendency (still within factors of 2 and 5) to measure lower INP concentrations over the full temperature range as compared to INSEKT. Recall that the flows for the FRIDGE filter collection were associated with a higher degree of uncertainty due to a miscalibration of the flows and the occurrence of a leak (see section 2.3.1), which might have caused this difference. In addition, the methods use different suspension volumes for INP detection. However, measurements with INSEKT and FRIDGE at the Jungfraujoch show a good agreement (Lacher et al., 2021), which indicates that the larger uncertainty in the present study was not caused by the different suspension volumes, but rather arises from the larger uncertainty in the sample flow from FRIDGE.

As shown in Fig. 8b, the UNAM-MOUDI-DFT tends to measure higher INP concentrations compared to INSEKT. This bias may be coming from the method used to capture the particles. While for the INSEKT samples Nuclepore filters were used, in the UNAM-MOUDI-DFT particles were impacted on glass coverslips. A possible explanation is that not all particles are released from the Nuclepore filters. If so, this may relate to the aerosols sampled at Puy de Dôme, as this bias was not seen in some prior comparisons (e.g., Mason et al., 2015). Moreover, the UNAM-MOUDI-DFT is the method using the fastest cooling rates of 10 °C per minute, such that an effect of a time dependency of ice nucleation might have impacted the results (e.g., Hoose and Möhler, 2012; Budke and Koop, 2015). However, this would have led to an underestimation of INP concentration, such that we conclude that the ambient INP concentration is not considerably controlled by stochastic variation, or that other instrumental properties of sample collection and analysis with UNAM-MOUDI-DFT are dominant.

Again, the IS and LINDA, sampling filters on the rooftop, tend to measure higher INP concentrations (Fig. 8e, f), and only 27% and 19% are within a factor of 2 of the INSEKT measurements, respectively. As INSEKT is a re-built of IS, a difference due to their setup is unlikely. A possible explanation is that filter measurements for offline INP analysis using standard inlet systems could systematically lose aerosol particles which are crucial for INP measurements. This could be supermicron particles, that are lost by impaction in bends or might not be sampled especially under high-wind conditions, and nanometer-sized particles that are lost by diffusion. The ability of nanoparticles to nucleate ice is not well investigated but it is suggested that pollen particles can release ice-active nanoscale particles (Duan et al., 2023). Larger particles are often associated with dust or Pollen, which are known to be efficient INPs (e.g., Murray et al., 2012). Calculations of the size-dependent inlet transmission efficiency indicate that the majority (84%) of 10 µm particles were still sampled via the WAI at a wind speed of 10 m s[-1] and

63 % at a wind speed of 15 m s$^{-1}$ (Hangal and Willeke, 1990), which is an upper value measured during the campaign.

To investigate this further, IS sample suspensions were size-segregated (Fig. 9) for three cases when there was a discrepancy to the measurements at the WAI (12[th], 14[th], 15[th] daytime, Fig. 9a, b, c) and one case when there was a good agreement (16[th] daytime, Fig. 9d). Those experiments reveal that the ice nucleation efficiency was not reduced significantly by filtering particles to < 3 and < 0.8 µm within the measurement uncertainties. Only on the 12[th] and 16[th] of October did a difference between the unamended and filtered samples occur. In fact, in some cases, the filtered experiments reproduced the unamended results. This indicates that the discrepancy between rooftop and WAI samples does not only arise from a non-sampling of larger INPs, at least those remnants in liquid suspensions after the first freezing experiment was performed. Another source of discrepancy could be that fragmentation or disaggregation of especially larger particles when placed in suspensions leads to a high bias in INP concentrations, as discussed already by DeMott et al. (2017). Indeed, the open-face Nalgene sampler can sample larger particle fragments, which could release multiple aerosols once suspended in water.

Whether this is only an issue for ground-based sampling locations but not for aircraft measurements due to, for example, typical decreases in large particle concentrations with altitude, needs to be investigated in future studies.

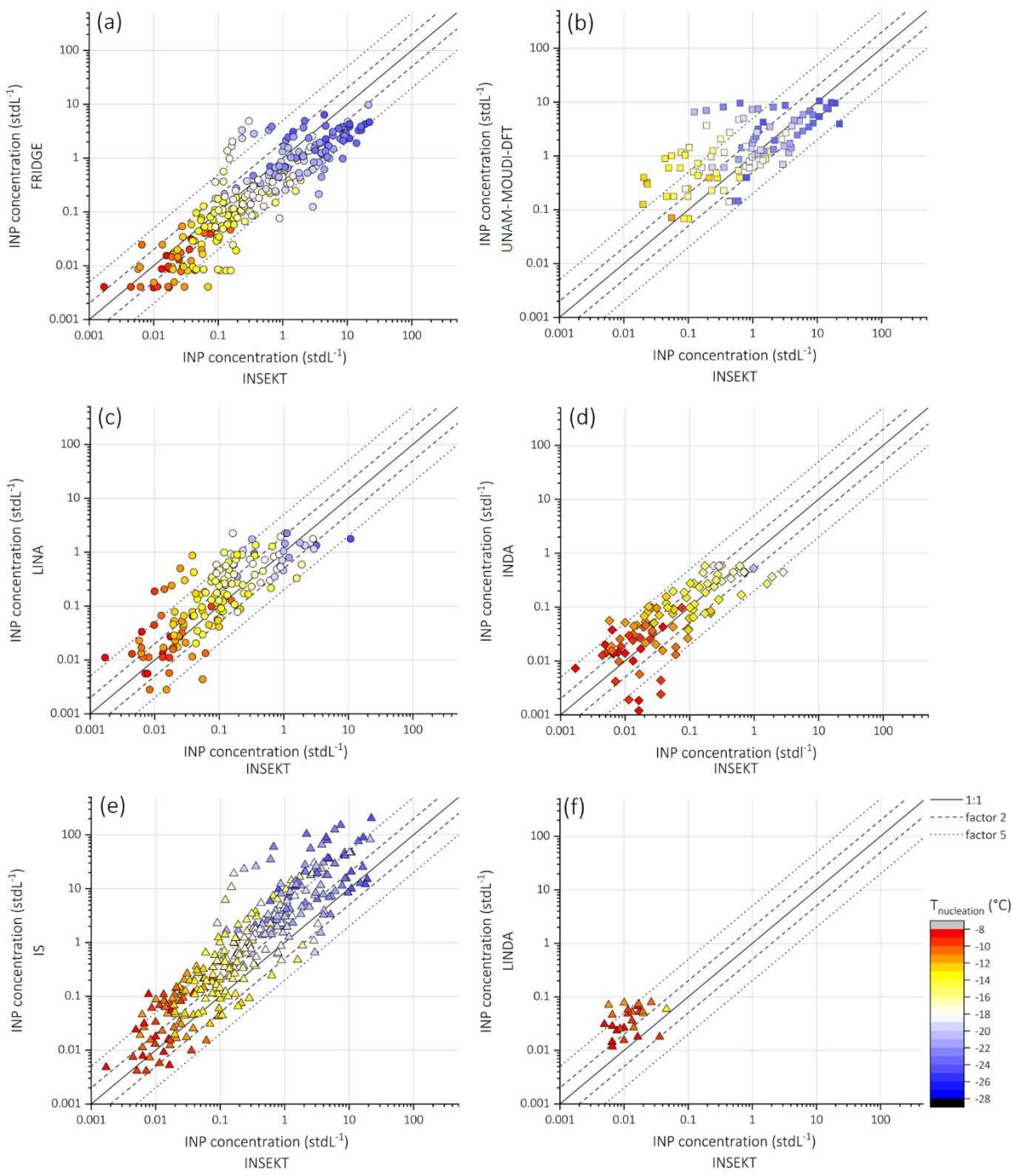

Figure 8: INP concentrations measured with FRIDGE (a), UNAM-MOUDI-DFT (b), LINA (c; standard filter holder), INDA (d; standard filter holder), IS (e; filter taken on rooftop), and LINDA (f; filters taken on the rooftop) as a function of INP concentrations measured with INSEKT; color-coding represents $T_{nucleation}$.

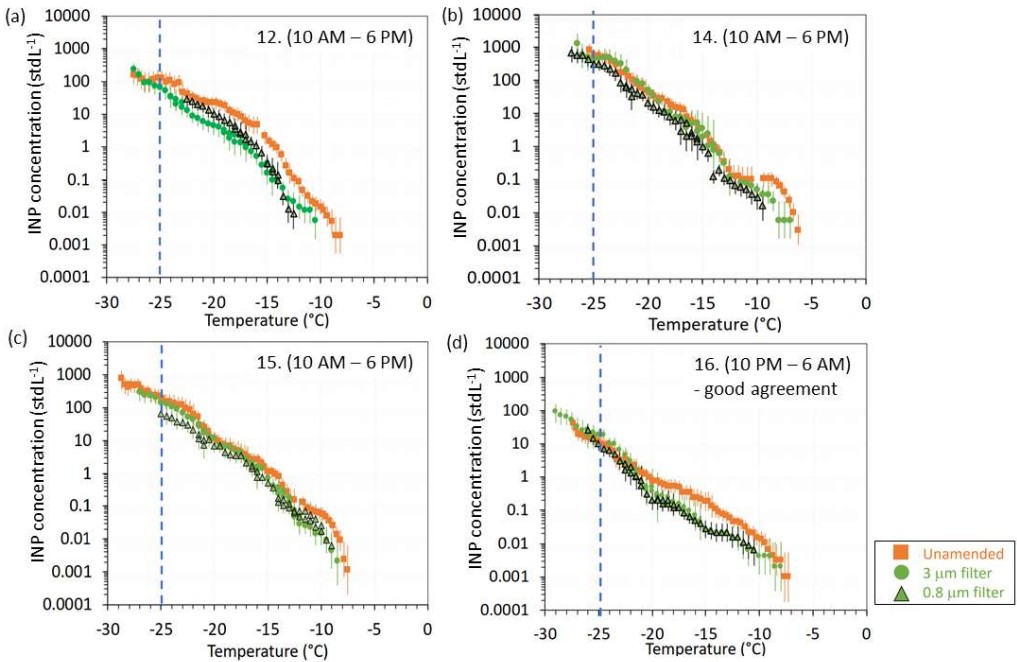

Figure 9: Size-segregated INP concentration as measured from filtering IS liquid suspensions; unamended freezing solutions (orange squares), solutions including particles < 3 µm (green circles) and < 0.8 µm (green triangles) are analyzed for the daytime sampling period of the 12th (a), 14th (b), 15th (c), and 16th October (d); only on the 16th October IS shows a good agreement to the INP concentration measurements at the WAI; blue line indicates the freezing temperature at -25 °C.

### 3.2.1 Investigation of INP differences using aerosol particle measurements

A wider spread between the methods based on filters collected at the rooftop and in the laboratory via the WAI is observed during many sampling intervals. In order to get a better insight into this deviation, the time series of the difference between the INP concentration measurements from the IS (rooftop) and INSEKT (laboratory) is investigated in relation to the wind velocity and the concentration of aerosol particles. Those freezing methods were selected as they are based on the same freezing analysis principle, and both span a large range of $T_{nucleation}$. As seen in Fig. 10, the difference between the INP concentration measurements from the IS and the INSEKT at -10 °C, -15 °C, and -20 °C, given as the lognormal difference, are sometimes occurring during elevated wind velocities (Fig. 10b), which can decrease the transmission efficiency, especially of larger particles, as discussed earlier. No relation between the difference of IS and INSEKT is observed to the total particle number concentration, and the particle number concentration 0.1 – 0.5 µm (Fig. 10c, d). Moreover, a higher ratio between IS and INSEKT is not observed during times of higher concentrations of particles between 0.5 and 2.5 µm and 1 and 2.5 µm (Fig. 10d), which would have been an indication for a generally higher concentration of larger particles in the ambient air, and which might be preferentially lost in the inlet prior to the INSEKT filter samples. At the same time, all the aerosol particle concentration measurements (total, 0.1 – 0.5, 0.5 – 2.5, 1 – 2.5 µm) are especially higher in the second period of the campaign, starting from the 13th of October, when a higher discrepancy between IS and INSEKT is observed. This might indicate that the aerosol population changed, and could have caused this discrepancy, e.g., by an increased presence of larger particles that are not sampled at the WAI, or could have caused particle fragmentation in IS. This potential cause of discrepancy depends on the assumption that especially

the larger fraction of the aerosol particle population dominated the INP population. A study performed at the same location using a stage impactor for size-segregated measurements indeed revealed that INPs are mostly super-micrometer particles (Bras et al., 2022). It should be noted that the size distribution measurements were conducted at the WAI, thus, the interpretation of the presented time series of aerosol particles during those high-wind velocity times is limited. In order to precisely identify such an impact, more intensive measurements need to be conducted by, e.g., having aerosol particle size distribution measurements at the rooftop and in the laboratory simultaneously.

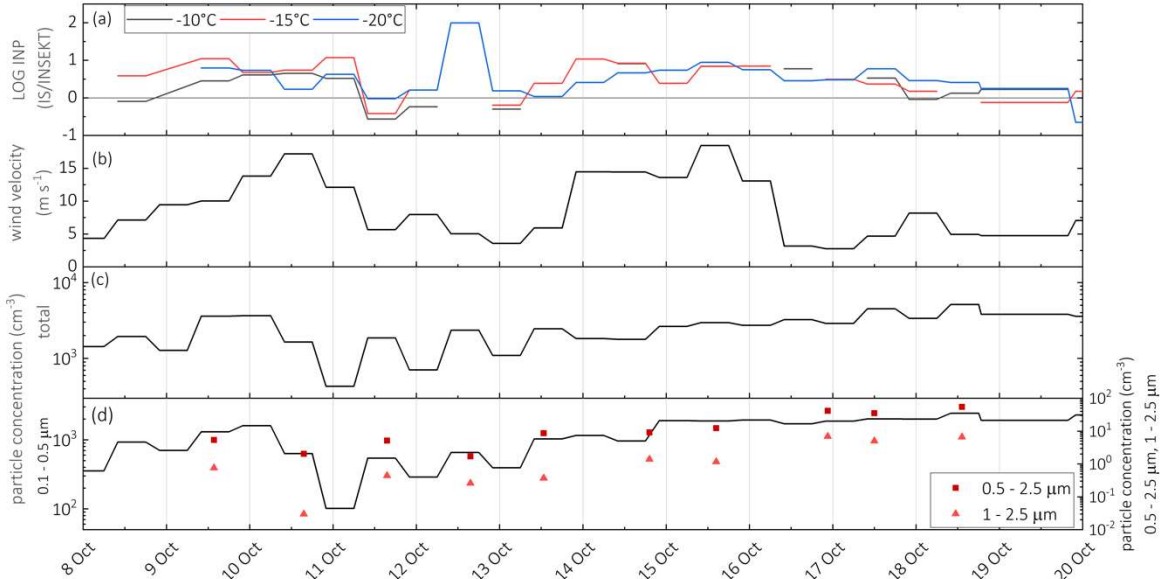

Figure 10: Time series of INP concentration differences between the IS and INSEKT at nucleation temperatures of -10 °C, -15 °C, and -20 °C (a), wind velocity (b), total particle concentration (c), particle concentration in the size range 0.1 – 0.5 µm, 0.5 – 2.5 µm, and 1 – 2.5 µm (d).

### 3.2.2 Comparison of INP concentrations using quartz fiber and polycarbonate filters

A subsample of the datasets was designed to test the possible influence of using different filter materials (quartz fiber versus polycarbonate filters). For this comparison, HERA and the standard sampler from TROPOS were operated in parallel using different filter materials. For the analysis, INDA and LINA, both operated at TROPOS, were used for evaluation. For the comparison shown here, HERA was equipped with polycarbonate filters (200 nm pore diameter), and the standard sampler with the quartz filter. Figure 11 shows results from sampling intervals between the 9[th] (daytime) to the 11[th] (daytime) of October. While both LINA and INDA can analyze particles collected with polycarbonate filters (creation of solution using the washing water), only INDA can analyze quartz fiber filter punches that are immersed in ultra-clean water. No systematic difference between the INP concentrations using those different filter materials is observed, showing a good agreement between INDA and LINA as previously reported (e.g., Knackstedt et al., 2018; Hartmann et al., 2019, Gong et al., 2020), which gives confidence that both materials can be used within the processing temperature ranges shown (≥ -20 ºC).

Moreover, quartz fiber filters and polycarbonate filters with different pore sizes (800 nm) were used simultaneously in the TROPOS standard filter holder and in HERA for the analysis with INDA and LINA during some sampling intervals. Quartz fiber filters were used from the 14th night-time (Fig. 6h) to the 16th daytime sample (Fig. 7, a - c), and 800 nm polycarbonate filters for the sampling intervals from the 16th (night-time) to the 18th (daytime; Fig. 7, d - g). When comparing with the overall INP measurements from the other methods, there was no noticeable influence of using quartz fiber filters, or polycarbonate filters with 800 nm pores, as compared to measurements using Nuclepore filters with a pore size of 200 nm. This shows that filters with a pore size of 800 nm and applied flow rate still have a sufficiently high collection efficiency for the majority of atmospheric INPs present during the PICNIC study. This is in agreement with Soo et al. (2016), who examined the collection efficiencies of a range of different filter materials and pore sizes for test particles with rather small sizes between 10 and 412 nm. They reported that the collection efficiency for polycarbonate filters with 800 nm pore sizes and the flow rates used here ( > 11 LPM) are above 97% for all particles in the examined size range (10 – 412 nm).

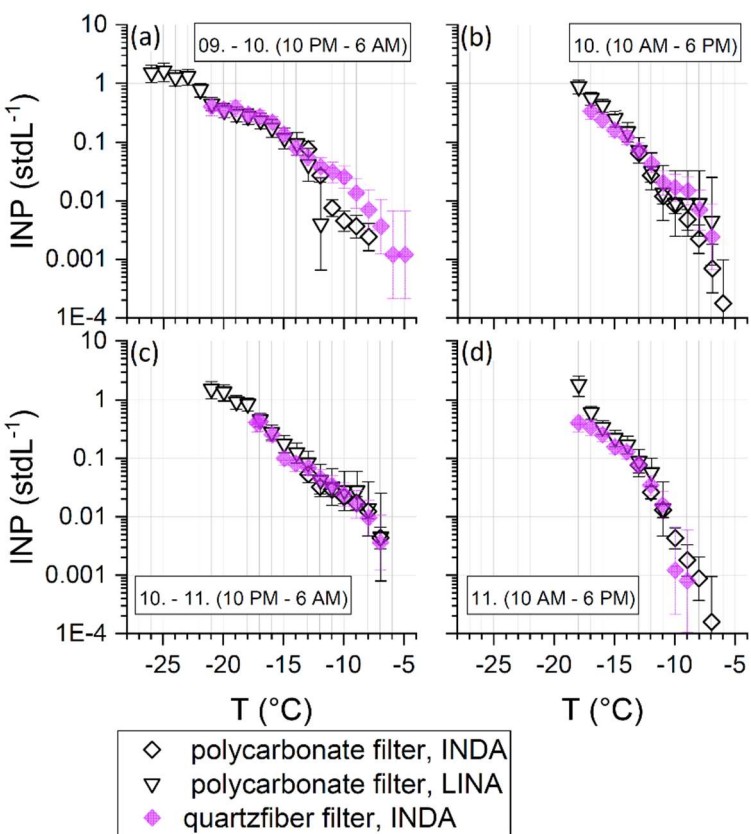

Figure 11: Comparison of different filter materials for parallel collected filters using INDA and LINA.

### 3.3 Comparison of online and offline methods

The comparisons presented in Figs. 5, 6, and 7 also include the measurements obtained from the CSU-CFDC, SPIN, and PINE. They are measured within the same time period of the filter collection but represent the instruments' specific time resolution, which is ~ 1 minute for CSU-CFDC, and ~ 10 minutes for SPIN and PINE. The measurements with PINE cover the full 8-hour filter collection time with a few exceptions.

Generally, the INP concentrations from the online instruments compare well with the offline techniques and are within the range of the offline-determined INP concentrations measured at the WAI. There is a slight tendency for the online instruments to measure lower INP concentrations, especially on the 10[th] of October (day- and night-time; Fig. 5g, h). This low bias might be explained by the limitations of the instruments to measure only particles below 2.5 µm by the use of impactors (CFDCs), or below 4 µm due to the natural loss in the tubes for the PINE instruments (Möhler et al., 2021). Thus, it might be possible that the filters used for the offline INP analysis sampled a higher fraction of larger aerosol particles that were ice-active. Moreover, the good agreement of online and offline INP measurements at the WAI indicates that the potential disaggregation of aerosol particles into many INPs in liquid solutions via the bulk immersion freezing techniques is not of major importance, at least for the measured size distribution at the WAI.

In general, the measurements from the online INP instruments reveal that INP concentrations at a given temperature vary up to an order of magnitude during the sampling interval of 8 hours, a variability that cannot be detected by the offline methods. A combination of both online and offline techniques is therefore of great advantage to capture both the INP concentration over a wide temperature range and their variability at single temperatures.

### 4 Summary and conclusion

During the PICNIC campaign in October 2018, a suite of online and offline INP measurement techniques was operated simultaneously to compare the temperature-dependent INP concentrations relevant to the formation of mixed-phase clouds. The methods were deployed in their typical operation configuration without equalizing measurement setups. Two CFDCs (CSU-CFDC and SPIN) and an expansion chamber (PINE) measured INP concentrations in the temperature range from -20 °C to -30 °C. INP concentrations were compared within ± 10 minutes and ± 1 °C to ensure that sampling and nucleation conditions were as close as possible. PINE agreed well with CSU-CFDC and most INP concentration measurements were within a factor of 2 (71%). During the cloud formation process in PINE, it is conceivable that not all aerosol particles are activating into cloud droplets during the expansion-induced cooling process, which can cause a low bias of immersion freezing INPs. Also, in CFDCs, it is possible that not all aerosol particles under investigation are exposed to targeted supersaturation conditions due to aerosol spreading beyond the aerosol lamina (DeMott et al., 2015; Garimella et al., 2017). Indeed, the comparison of CSU-CFDC and SPIN reveals that SPIN measured lower INP concentrations (only 35% of the data are within a factor of 2), which could arise from different degrees of aerosol spreading beyond the lamina. The supersaturation was lower in SPIN (2.8 ± 1.9%) as in CSU-CFDC (6.5 ± 1.4%) and the instrument-specific size threshold to identify ice crystals was larger in SPIN (5 µm) as in CSU-CFDC (4 µm). Therefore, it is conceivable that fewer particles in SPIN were activated into cloud droplets and ice crystals, or they were not growing to ice crystals large enough to be classified as ice. More specific tests to characterize the effect of aerosol

spreading beyond the lamina during field studies, as well as laboratory characterization of the established supersaturation conditions, and hence cloud droplet and ice crystal activation, should be performed in future

studies. More such intensive INP intercomparisons, resulting in a larger dataset, should be conducted in the future to better understand discrepancies amongst the online instruments and to guide potential technical mitigations.

INP filter sampling was performed during day- and night-time for 8 hours and analyzed with FRIDGE, INDA, IS, INSEKT, LINA, LINDA, and UNAM-MOUDI-DFT. The filters for IS and LINDA were collected directly in ambient air on the rooftop of the laboratory, while the other filters were collected behind the WAI in the laboratory.

The methods using filters collected at the WAI generally show good agreement over the investigated temperature range when compared to INSEKT as a reference, as > 45 % are within a factor of 2. This indicates that, with attention to protocols for filter handling and analysis, not only the different freezing procedures (droplet freezing, freezing of suspensions) but also the sampling devices (standard filter holders, FRIDGE custom-built semi-automated sampler, open-faced Nalgene units, MOUDI, HERA) and sampling substrates (PTFE fluoropore

membrane, quartz filters, hydrophobic glass coverslips, polycarbonate filters (200 and 800 nm pore diameters)) can be used together to provide generally consistent and reliable measurements of INP concentrations. It should be pointed out that the faster cooling rate (10 °C min$^{-1}$) of the UNAM-MOUDI-DFT did not lead to lower INP concentrations as compared to the other methods, indicating that the time-dependence of nucleation is of secondary importance for immersion freezing on ambient particles in this study. The IS and LINDA sometimes measured

higher INP concentrations, and as compared to the INSEKT method, only 27 and 19% of the data derived with IS and LINDA are within a factor of 2, respectively. This occurred sometimes during high-wind conditions and might be explained by losses of super-micrometer aerosol particles and INPs in the WAI. Calculations of particle transmission efficiencies reveal that the majority (>90%) of 10 μm particles are sampled at the WAI. Next to this potential non-sampling of larger aerosol particles, it is also conceivable that in-suspension

fragmentation/disaggregation of especially larger particles, which were more often sampled on the rooftop, results in an elevated INP concentration, as discussed in DeMott et al. (2017). It should be noted that such a fragmentation leads to an artificially high INP concentration, as the initial particle would only lead to the freezing of one cloud droplet in which it is immersed. Moreover, most ambient INP measurements are performed behind aerosol inlets, and a systematic undercounting or overestimation should be investigated in future studies. For example, the aerosol

particle transmission efficiency should be measured during different sampling conditions with regard to meteorology and the presence of particles in the size range relevant to ice nucleation. Moreover, specific experiments for a potential particle fragmentation and increase in INP number concentration should be conducted by measuring aerosol particles and INPs impacted and directly counted on a substrate and after re-suspending the impacted aerosol in solution. In addition, different rooftop configurations could be tested in parallel using no inlet,

and different PM inlets (e.g., PM10, PM2.5).

The INP measurements of the online instruments, that were performed within the same sampling intervals of the filter collection time, agreed well with the results from the offline methods. The online instruments showed a slight tendency to measure lower INP concentrations during some sampling intervals, which might be caused by the restriction of the online instruments for sampling aerosol particles smaller than 2.5 μm, which is needed to avoid

the misclassification of unactivated aerosol particles as ice crystals. Nevertheless, we conclude that the presented methods here are suitable for combination with offline methods, which is required in order to capture the complete temperature range relevant for heterogeneous nucleation in the mixed-phase cloud regime. In addition, based on

the finding of a good agreement between online and offline methods at the WAI, we conclude that the potential breakup of aerosol particles, that pass through the WAI, into many INPs via the bulk immersion freezing technique is of minor importance for particles below approximately 10 µm.

With regard to required precision of INP measurements to be included in models, the results from our study greatly demonstrate that the methods used in their original configuration agreed overall well within a factor of 5.

Especially in light of ongoing efforts for INP monitoring networks, we recommend that such intensive INP intercomparison measurements are repeated frequently, during different seasons, and at measurement sites characterized by different aerosol particle sources and properties. Ambient INP intercomparison campaigns are useful in addition to laboratory campaigns, where specific aerosol particles are used as test material. Such efforts are needed to ensure accurate INP concentration measurements, which is required to better understand and represent INPs in the atmospheric system.

## Data availability

*The data used in this study will be made available via the KITopen data repository.*

## Author contribution

LL wrote the manuscript with contributions from YB, PJD, LAL, CRR, JS, HW, MW, and EF. CSU-CFDC measurements and analysis were provided by KB, PJD, EJTL, and KAM. DJC, MG, and MW conducted the SPIN measurements and analysis. PINE measurements were performed by LL, MA, and OM, and CB, NB, RF, BJM, JN, TP contributed to the instrument setup and data analysis. HB, DC, SR, JS, and ET performed the FRIDGE filter measurements and data analysis. Filter sampling for INDA and LINA and data analysis were conducted by CJ, SM, FS, and HW. KB, PJD, TCJH, and EJTL performed and analyzed the IS measurements. INSEKT measurements were performed and analyzed by BB, LL, KH, and OM. YB and EF provided the measurements from LINDA. UNAM-MOUDI-DFT measurements and analysis were conducted by LAL and CRR. DP, KS. MR was responsible for the logistics. EF provided the meteorological measurements at the Puy de Dôme station and coordinated the PICNIC campaign.

## Acknowledgment

We thank the technical team from the Puy de Dôme / OPGC (l'Observatoire de Physique du Globe) for support and service during the campaign. We acknowledge the KIT technical team with special thanks to Steffen Vogt.

## Competing interests

The authors declare that they have no conflict of interest. Luis A. Ladino and Ottmar Möhler are members of the editorial board of Atmospheric Chemistry and Physics.

## Financial support

This research received funding from the European Commission under the Horizon 2020 –Research and Innovation Framework Programme via the ACTRIS-2 Trans-National Access, and from the ANR-CHAIN project number ANR-14-CE01-0003 -01. Larissa Lacher received funding from the KIT Technology Transfer Project N059. Barbara Bertozzi acknowledges funding from the European Union´s Horizon 2020 research and innovation program under the Marie Skłodowska-Curie grant agreement No 764991. Martin Wolf and Daniel Cziczo acknowledge funding from the US National Science Foundation grant AGS-1838429, a supplement to Collaborative Research: A Closure Study of Mixed Phase Clouds at Storm Peak (AGS-1749851). Colorado State University co-authors received partial funding support from U.S. National Science Foundation Award No. 1660486. Luis Ladino and Carolina Ramirez acknowledge partial funding from Conacyt through the CB-285023 grant. Conrad Jentzsch and Stephan Mertes were supported by the German Research Foundation (DFG) in SPP 1294 under grant no. 316508271 and received funding for the campaign from the ACTRIS-2 Trans-National

Access. Stephan Mertes also received funding from the German Research Foundation (DFG) – Project Number 268020496 – TRR 172, within the Transregional Collaborative Research Center "ArctiC Amplification: Climate Relevant Atmospheric and SurfaCe Processes, and Feedback Mechanisms (AC)3." Erik Thomson and Dimitri Castarede have been supported by the Swedish Research Councils, VR (2013-05153, 2020-03497) and FORMAS (2017-00564), and by the Swedish Strategic Research Area MERGE. Benjamin J. Murray and Mike Adams received funding from the European Research Council grant 648661 MarineIce. The authors gratefully acknowledge CNRS-INSU for supporting measurements performed at the SI-COPDD, and those within the long-term monitoring aerosol program SNO-CLAP, both of which are components of the ACTRIS French Research Infrastructure, and whose data is hosted at the AERIS data center (https://www.aeris-data.fr/)."

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
