# Peer review of "The Puy de Dôme ICe Nucleation Intercomparison Campaign (PICNIC): Comparison between online and offline methods in ambient air"

_EGUsphere, 2023_

## Author Comment (AC1)

**Answer to Reviewer 1**

We thank the reviewer for his/her helpful and detailed comments and suggestions, and believe that with the corrections made our manuscript has improved significantly.

The reviewer's comments are in blue, and our answers in black. Sections from the original manuscript are presented in *black italic* and corrections in *red italic*.

In this paper, the authors present the ambient ice-nucleating particle (INP) number concentration obtained using online and offline measurement techniques during PICNIC campaign at Puy de Dôme. Intercomparison between online and offline instrumentation, as well as the impact of sampling site and sampling setups for offline instruments were assessed. The authors addressed the necessity of online and offline INP measurement instruments nicely. Such instrument intercomparison is essential and of great significance for the ice nucleation and atmospheric community and requires a lot of effort. Therefore, the paper fits the scope of ACP.

However, the quality of the paper should be improved before acceptance for publication in ACP. There are several typos and inconsistent usage of abbreviations in the manuscript. The reviewer tried to go from line to line to edit the manuscript, but the authors hold the responsibility for a thorough typo, format, and grammar check before re-submission.

Major comments

1. Residence time is a critical factor that can affect online INP concentration measurement. The discussion of different residence times for different online instruments requires elaboration.

The reviewer is right that a discussion about the difference in residence time is missing in the manuscript. While the residence time of the CSU-CFDC is mentioned in section 2.2.1

*„The residence times of aerosols in the supersaturated region are 5 s for the flow rate used (1.5 LPM; liter per minute)."*

, we add the residence time in SPIN to section 2.2.2

*„Aerosols are fed into the chamber at a sampling rate of 1 LPM and constrained to a lamina center-line with 9 LPM of sheath air. The residence time of the particles in the chamber is 10 seconds."*

And a discussion related to the ice crystal size in the same section:

*„Although SPIN is operated at a lower supersaturation as compared to the CSU-CFDC, the ice crystal have a longer residence time (10 seconds) such that they grow to sizes larger than 5 µm."*

, and included the following description about the residence time in PINE in section 2.2.3

*„ In the PINE instrument, the residence time of aerosol particles at supersaturated conditions or in supercooled droplets is more variable as compared to CFDC instruments. The time during which cloud droplets are present during an expansion is 33 seconds. However, it should be noted that this is an upper limit for the residence time, as ice crystals formed by INPs are detected during the whole expansion period and each INP has its own trajectory within the cloud chamber."*

Moreover, we add the following discussion about the difference in residence time to section 3.1 (*Intercomparison of online instruments*):

*„Please note that the residence time of SPIN is longer (10 seconds) as compared to CSU-CFDC (5 seconds), however, we believe that other factors such as the difference in supersaturation are more important here."*

*„In addition, the residence time of particles in PINE is not as well quantified as in the CSU-CFDC, and might be longer (maximum 33 seconds), that might impact INP concentrations."*

2. The logical flow of the introduction section needs to be more organized.

We hope that the revised introduction is more organized.

3. The presentation quality, for example, marker colors in figures, should be reconsidered.

As the reviewer is not specifically pointing out which figure or table he/she is referring to, we would not like to make changes. With many instruments and cross-comparisons, we have done our best to clearly distinguish measurements using point types and colors.

4. The mixed-use of units and abbreviations is problematic.

By addressing your and the second's reviewer comments regarding this, and by carefully checking the manuscript, we hope that all abbreviations and units are used properly now.

Specific comments

L4: ice nucleating particles -> ice-nucleating particles.

Corrected.

L5: It would be more informative for the readers if the authors could provide numerical ranges of the temperature range and the orders of magnitude here.

We include the temperature range now, but the orders of magnitude were already given.

*„…the relevant temperature range for mixed-phase clouds (< -38 °C) covers up to ten orders of magnitude,…"*

L10: What are the wall temperatures of the two CFDCs in immersion freezing mode at -5 °C?

We corrected the sentence, as no online INP measurements were performed at temperatures warmer than ~ -20°C

*„INP concentrations were detected  in the immersion freezing mode, between ~ -5 °C and -30 °C."*

L13: "temperatures below -20 °C" contradicts with the statement in L10.

Agreed and changed (see comment above).

L16: Missing a comma before INDA.

Corrected.

L22: It's better to clarify these are temperature spectra.

Corrected.

*„…and the obtained INP freezing spectra were compared at 1 °C steps. …"*

L23: Not all offline instruments were operated at 1 °C step according to Table 1.

The cooling rate (temperature change per minute) is not the temperature interval at which the instruments are compared; the cooling rate describes the change in temperature during the cool-down process. Indeed, not all the instruments were operated at a cooling rate of 1 °C per minute, as described in Table 2 (formerly Table 1), however, the results of the different instruments are compared at a 1°C step.

L25-27: The explanation of the discrepancy between SPIN and CSU-CFDC is confusing. CSU-CFDC is also a CFDC-type instrument, isn't it?

Yes, both the instruments are CFDCs. We are explaining this discrepancy in more detail in the results and discussion, as we wanted to keep the abstract as short as possible.

L30-31: The description of the offline sampling technique adds little to the results. Either combine it with the description between L13-15 or remove it.

We deleted the sub-sentence in lines 30 – 31.

L33-34: Did the authors collect filter samples simultaneously on the rooftop and in the laboratory? Fig. 1 shows that IS and LINDA analyze filters from the rooftop, and the rest five offline instruments analyze filters collected in the laboratory.

Yes, those measurements were all performed simultaneously, as written in lines 21 and 22. We also add to line 34 now:

*„… compared to measurements performed simultaneously behind the whole air inlet system."*

L36: Do the authors mean primary ice formation?

Yes, *„The first formation of ice…"* refers to primary ice formation.

L53-55: Could shorter sampling time for offline measurement techniques result in a smaller sampling volume, and therefore INP concentration below the detection limit?

Yes, the same sample flow with a shorter sampling leads to a lower or even zero number of very rare INPs on the same filter; we include this statement now:

*„Results from offline INP measurements can also be obtained for shorter periods, however, this impacts the limit of detection and may lead to a lower or even zero number of very rare INPs."*

L58: By saying organic INPs, do the authors mean biological INPs here? It would be beneficial for the readers if the authors can add a few lines here to briefly discuss the sampling size limit of most online instruments, which normally excludes pollens and dust particles above 10 - 20 μm that are ice-active at warmer temperatures and could be captured by offline sampling.

Organic refers to biological and biogenic material. We discuss the discrepancy caused by an incomplete sampling of online and offline techniques detailed in lines 73 – 103.

L70-72: Please reword.

We changed the sentence to:

*"Below -10 °C, instruments showed good agreement using SNOMAX® and natural dust samples. Discrepancies occurred using SNOMAX® above -10 °C, with illite NX above -25 °C, and with potassium feldspar between -20 and -25 °C."*

L74-87: The logic can be improved here. The statement here is a general comparison between the size ranges of online and offline sampling techniques, instead of "the size range of aerosol particles that are INPs". These lines should be combined with the paragraph before, please refer to the previous comment on L58. Following the impact of aerosol type and nucleation temperature in instruments' comparability at the end of the last paragraph, a review of previous intercomparison results showing the impact of aerosol size range on INP concentration measured by different instruments is missing here, which is helpful for the readers.

INP can have different sizes and it is not trivial to infer them, especially in ambient air. As we do not infer the size of the INP in this study, we would like to only give this general statement about the sampling size range of the online and offline techniques, and the general size range of the dominant INPs.

We also reference other intercomparsion studies in the laboratory and the field controlling the size range of aerosol particles now:

*„In laboratory-based intercomparison studies, it was suggested that generally good agreement between methods was achieved by controlling the aerosol particle size distributions used for the INP experiments (Wex et al., 2015; DeMott et al., 2018; Burkert-Kohn et al., 2017). At ambient conditions, however, aerosol particles and INPs can span a wide size range, which can be crucial for determining the real ambient INP concentration, and for comparing INP measurement techniques that cover different size ranges (Knopf et al., 2018).*

L88: a -> an.

Corrected.

L102-103: Is this statement relevant to instrument comparison?

In the light of the potential non-sampling of large INPs using online instruments, it is, in our opinion, an important aspect of intercomparison studies in ambient air, in close proximity or further away from emission sources of large particles.

L104-107: Consider replacing "Moreover". Is this paragraph relevant to instrument comparison? If yes, please organize the logic.

We rephrased the paragraph to:

*"Ambient INP concentrations can be close to typical instrument detection limits (Boose et al., 2016a) and the way measurements close to detection limits are considered for averaging INP concentration over longer sampling intervals, which can be done for comparing different instruments, is another important aspect of making ambient measurements. Ambient INPs show a wide range of concentration across the relevant temperature range (e.g., Kanji et al., 2017), and it should be ensured that even low numbers of INPs, close to instruments' detection limits, are captured."*

L114: Move the definition of CSU-CFDC to L111 after the first appearance of "online instrument".

Done.

L115: Please reword.

We changed the sentence to:

*"However, a high bias for offline methods, sampling particles onto filters or into a bulk liquid, against an online method was observed below -20 °C."*

L130: What types of cloud form and occur at Puy de Dôme? Is it liquid clouds or mixed-phase clouds? If they are liquid clouds, how are the aerosols connected to mixed-phase clouds?

Both liquid and mixed-phase clouds occur at Puy de Dôme, however, the cloud phase was not determined during the PICNIC campaign. Mountainous regions typically act as a pump for air masses into more elevated levels of the atmosphere, thus our measurements of INPs at the Puy de Dôme station are seen as a potential INP population that can be lifted higher up in the atmosphere.

L136: was -> were.

Corrected.

L138-139: Consider removing the statement.

To point out differences as compared to other studies, we would like to keep this statement.

L150-151: Please specify the size range of SMPS. Why do the authors couple an SMPS to a CPC?

The size range of the SMPS is 10 to 560 nm, we include the size range now in the text. As this is a custom-made SMPS, we mention the CPC model used to measure the number concentration of the size-selected particles. We re-formulated the sentence to:

*„The submicron aerosol particle size distribution was measured using a custom-made scanning mobility particle sizer (with a particle diameter range from 10 – 560 nm) operated with a condensation particles counter (CPC, model 3010, TSI)"*

L155-156: Please elaborate on the characterization of WAI. How was the transmission efficiency computed, using number or mass concentration? What instrument was used to measure the concentration of 10 µm particles? Can this explain the consistently higher INP concentration from the filters sampled from the rooftop? Did the authors measure filters collected on the rooftop and downstream of the WAI during the same period with the same instrument and check the difference?

The transmission efficiency of the WAI is inferred by calculations, no measurements were performed for characterizing it. We include now the references for those calculations:

*„Moreover, the transmission efficiency of the WAI is dependent on wind speed. Calculations show that at values of 7 (10) m s-1, 93% (84%) of the particles with a diameter of 10 µm are entering the inlet (Hangal and Willeke, 1990; Baron and Willeke, 2002)."*

As discussed later in our text, this might explain discrepancies, in combination with a breakup of larger aerosol particles within liquid suspensions.

*„Calculations of particle transmission efficiencies reveal that the majority (>90%) of 10 µm particles are sampled at the WAI. Next to this potential non-sampling of larger aerosol particles, it is also conceivable that in-suspension fragmentation/disaggregation of especially larger particles, which were more often sampled on the rooftop, results in an elevated INP concentration, as discussed in DeMott et al. (2017)"*

We sampled filters at the rooftop and at the WAI simultaneously and analyzed them with IS (rooftop) and INSEKT (WAI). As INSEKT is a re-built of IS, we concluded that differences do not mainly come from instrumental differences.

L157: Delete ", and".

Done.

L180: The data points are already very limited (only 20 and 34 points according to Table 2) for a 14-day measurement period with a time resolution of 10 min. How could this be?

First, this is because the data from the online INP instruments were compared at temperature intervals of ± 1 °C and overlapping sampling time. As the CFDCs are not automated and continuous operating instruments, daily sampling times were approximately two times 4 hours; each sampling period is thereby 10 minutes between two 5-minute filter background measurements. Second, for some time during the day, SPIN was operated at colder temperatures, relevant for cirrus clouds (Wolf et al., 2020). Moreover, as stated in the manuscript, due to a temperature calibration of PINE after the campaign, fewer data points were available at the temperature chosen for the intercomparison.

*„It should also be pointed that, due to a temperature calibration performed after the PICNIC campaign, the PINE had fewer overlapping measurements with CSU-CFDC as initially targeted."*

L181-183: What is the flow rate of PFPC in this study? What type of impactor(s) was used?

The flow rates were the same as in Gute et al. (2019), as well as the impactor, which is part of the instrument. We include this now in the text:

*„The PFPC was deployed at a separate inlet and used an impactor with a 50% size cut at 2.5 μm. The inlet and outlet flow of the PFPC were kept at the same values as described by Gute et al. (2019), i.e., 250 LPM and 10 LPM, respectively.“*

L186-192: What are the INP concentration factors for the other two online instruments? These values should be reported even though the authors decided to use 11.4 for all three instruments. Do different geometries and structures of the online instruments, as well as the impactors installed at instrument inlets affect INP concentration factors? An assessment is needed.

As SPIN is operated with a 2.5 μm impactor, no major difference to the CSU-CFDC is expected during times sampling and not sampling at the concentrator. PINE is operated without an impactor when sampling not at the PFPC, such that variability might arise in the INP concentration factor. We include now the following statement:

*„For the intercomparison between the online INP instruments, the same INP concentration factors were applied for simultaneous measurements. This did not have an impact on the instruments' comparability, given that the instruments did not use additional impactors smaller than the PFPC's impactor with a size cut of 2.5 μm. The INP concentration factor used for the online intercomparison is thereby a campaign average of 11.4 and has a standard deviation of 1.7. This INP concentration factor was inferred by consecutive measurements with the concentrator turned on and off sequentially, using CSU-CFDC, which performed such measurements most frequently. The average concentration factor derived with PINE was similar (campaign average 10.9) but with a higher standard deviation (5.8), that might arise from the fact that PINE does not use an impactor when not sampling at the concentrator, such that larger particles, that are ice-active, can enter the instrument and contribute to more variation of the measured INP concentrations.“*

L206-208: Please specify the size range of the OPCs for the three online instruments. What's the transmission efficiency of the single-jet impactors? Do the impactors modify the sampling flow or aerosol population entering CSU-CFDC compared to the other two online instruments?

As the size range of the OPCs is a detail that is very instrument-specific, and do not impact the interpretation of the results, we do not see the need to state the size ranges of the OPC.

The CSU-CFDC has a 90% transmission efficiency from below the OPC lower size bin to the impactor size (DeMott et al. (2017), which is comparable to PINE (90%; Möhler et al., 2021) and SPIN (100% below 2 μm), thus we do not think that this impacts the comparability of the instruments.

As described in prior publications (e.g., Rogers et al., 2001; DeMott et al., 2017), diffusion dryers are used to reduce the RH in the sample air upstream of the impactors to immeasurable values (e.g., less than 5%), which implies a dew point temperature typically below -20 °C, preventing any chance for spurious supersaturations as the air cools to the steady state RH of the CFDC.

L240: Why do the lamina supersaturation and $RH_{water}$ have different uncertainties?

The uncertainties are the same. We have corrected this error.

*„The supersaturation employed for this study was 2.8 ± 2.5% (102.8% RH_water ± 2.5%)..."*

SPIN used diffusion dryer tubes filled with silica gel and molecular sieves upstream of the sample inlet. This brought the sample RH to below 20%.

L242: What's the transmission efficiency of the impactor? Will it change the sampling flow and aerosol population entering SPIN compared to the other two online instruments?

SPIN used a cyclone impactor with a D50 size threshold of 2.5 μm. Below 2 μm, the reported aerosol particle transmission efficiency is 100%. The use of the impactor should not change the sub-2 μm aerosol population sampled by SPIN as compared with that sampled by other online instrumentation.

INP concentrations measured with SPIN are directly compared to the CSU-CFDC, that uses an impactor with the same size cut, thus, we do not expect relevant discrepancies arising from this.

PINE is operated without an impactor and has a 50% transmission efficiency of particles < 4 μm, as stated in section 2.2.3, such that differences might arise.

We include such a discussion now at the end of section 3.1:

*„It should be noted that differences between the online instruments might arise from the difference in impactors. CSU-CFDC is operated with two single-jet 2.5 μm impactors, while SPIN is using only one, and PINE is operated without an impactor and thus has a 50% aerodynamic size-cut at 4 μm due to the loss of particles in its inlet."*

L252: Application of correction factor(s) may improve the systematic underestimation of SPIN compared to CSU-CFDC. Consider moving the clarification between L525-527 here.

We clarified our reasons for not using correction factors now:

*„As the degree of aerosol lamina spreading was not quantified in this study, no correction factor was applied."*

Corrected.

L258: Please keep consistent use of StdL or L in the paper, for example, L270, L274, and L293. Please check the figures and text throughout.

The description of the PINE instruments refers to volumetric flows (Möhler et al., 2021), e.g., during the expansion the volumetric flow is kept constant, thus the usage here of L is correct. Only when we refer to standard liters, we use StdL. E.g., the results of INP concentrations are given in stdL, in order to be able to compare our results to other measurements.

To clarify that all INP concentration measurements are reported in numbers per liter of air at standard conditions (stdL), we now added the following statements at the beginning of the online and offline sections:

*„Three different online INP instruments were operated behind the WAI in parallel for several hours per day. INP concentrations were determined for single particles activating in a temperature range between ~ -20 °C and  -30 °C, in the condensation/immersion freezing mode (via controlling processing relative humidity). All INP concentrations are referenced to standard liters sampled.“*

*„For offline INP analysis, aerosol particles were collected simultaneously with the different sampling setups during 8-hour intervals. ). All INP concentrations are given with reference to standard liters sampled.“*

L262: Please include "e.g." in the citation or complete the list of citations.

There is only this study about cirrus-relevant INP concentration measurements during the PICNIC campaign using SPIN. Thus, the usage of „e.g." would not be correct.

L292: welas-2500 **OPC**, delete "which".

Corrected.

L293: Specify the LOD is specific for the two consecutive experiments in this study.

*„The welas-2500 OPC has an optical detection volume of 10%, thus has a limit of detection of 2.5 INP per liter for two consecutive experiments.“*

L299: , however -> However. Please check the usage throughout.

Corrected.

L343-345: How long did the transport take? Why are the authors so certain such exposure to heat doesn't impact the results? What are the dominant INP source at Puy de Dôme?

Yes, the reviewer is correct, and we should be more cautious here. We rephrased this to:

*„The samples were not actively cooled during transport, however, given the relatively short travel time of ~ 8 hours to the laboratory in Frankfurt, we do not consider that this impacts the results, but cannot be excluded for certain (Beall et al., 2020).“*

No study to identify the dominant INP source at Puy de Dôme exists. A PINE instrument is installed permanently at the site now, such that in the future this question might be addressed in more detail.

L347: (2015 -> , 2015

Corrected.

L387: 100 nm deionized water?

Corrected to:

*„…two 100 nm - filtered deionized (DI) water rinses…"*

L415: Are there SEM images or other evidence to indicate "release of all particles from the filter"?

This is an interesting question and might be relevant for all filter-collection based samples. However, we did not perform such measurements for this ambient INP intercomparison study but are planning to perform these characterizing experiments for a separate study.

L426: Is "real insects" opposed to plastic or resin insects? "insects" would be sufficient.

We deleted „real".

L429: sec. -> Sect. Please refer to ACP guidelines.

Corrected.

L443: Does the camera illuminate the droplets?

Yes, the light below the tubes illuminates the tubes containing the solution.

L450: Define "UNAM".

We defined it now in the section title 2.3.7

*„2.3.7 Universidad Nacional Autónoma de México Micro-Orifice Uniform Deposit Impactor–Droplet Freezing Technique (UNAM-MOUDI-DFT))."*

and corrected the abbreviations in the text and figures.

L461: Please explicitly indicate aerodynamic particle size here.

Included now:

*„…determines the concentration of INPs as a function of temperature and aerodynamic particle size…"*

L465: x -> ×

Corrected.

L472: It's very nice to indicate the units of all physical quantities here. The unit for INP concentration is missing. Is $f_{nu}$ supposed to be dimensionless? What is the range of $f_{ne}$ used in this study?

The reviewer is right, the correction factors are dimensionless, which we indicate now in the manuscript:

*„… $f_{ne}$ is a correction factor, that varies between 1.2 and 4.7, and that takes into account the uncertainty associated with the number of nucleation events in each experiment"*

We believe that the usage here is correct.

We write now:

*"However, it should be noted that also the CSU-CFDC might not measure the total ambient INP concentration, due to aerosol lamina properties and size cuts, which will be discussed below in more detail and that can lead to an underestimation of the INP concentration."*

L499: Fig. 3, panel a -> Fig. 3a. Please replace other appearances accordingly.

Done.

L500-502: Could this be caused by the absence of correction factor(s)? Please refer to the comment on L252.

Indeed, we think that it is related to the aerosol lamina properties, which we write two lines after L500-502.

*"A possible explanation for this systematic deviation could be related to the aerosol lamina properties."*

We clarified our statement now:

*"… ultimately this may be an issue with how the central lamina is introduced to the chamber, and how the thermal gradients and non-laminar flow at the location where the aerosols are entering the chamber impact their spreading…".*

L515-523: What about the residence time difference for CSU-CFDC and SPIN?

The difference in lamina RH between the CFDC instruments makes an interpretation of any potential differences in INP number concentration due to differences in residence times convoluted.

We write now:

*"The impact of aerosol spreading was not quantified during the campaign, and data reported for the CSU-CFDC and SPIN instruments here remain original to account for this phenomenon"*

L528-529: Delete "(Fig. 3, panel b)". Add "Fig. 3" in the second bracket.

Done.

L532-535: What about the residence time of aerosols in CSU-CFDC and PINE?

Also the second reviewer was pointing out that we did not discuss the residence time of the particles in PINE, thus we added to the method section about PINE:

*„In the PINE instrument, the residence time of aerosol particles at supersaturated conditions or in supercooled droplets is more variable as compared to CFDC instruments. The time during which cloud droplets are present during an expansion is 33 seconds. However, it should be noted that this is an upper limit for the residence time, as ice crystals formed by INPs are detected during the whole expansion period and each INP has its own trajectory within the cloud chamber."*

And add the following discussion to the part of the manuscript the reviewer is referring to:

*„In addition, the residence time of particles in PINE is not as well quantified as in the CSU-CFDC, and might be longer (maximum 33 seconds), that might impact INP concentrations. ."*

L547-550: Again, is it possible to measure filters collected on the rooftop and downstream of the WAI during the same period with the same instrument and check the difference?

We did this using two identical setups: INSEKT (measuring at the WAI) is a re-built of IS (measuring at the rooftop). To point this out, we added the following sentence to the results comparing the other offline methods to INSEKT:

*„Again, the IS and LINDA, sampling filter on the rooftop, tend to measure higher INP concentrations (Fig. 8, panels e, f), and only 27% and 19% are within a factor of 2 of the INSEKT measurements, respectively. As INSEKT is a re-built of IS, a difference due to their setup is unlikely."*

Done.

L582-584: Please specify "a large temperature range". FRIDGE and IS seem to cover even wider temperature ranges in Figs. 5-7.

We included now the temperature range:

*„This method was chosen since filter collection for INSEKT was performed in the laboratory at the WAI inlet, similar to that for most of the other methods, and since it covers a large temperature range (approximately from -8 °C to -25°C) of INP measurement."*

We believe that INSEKT is the most suitable reference method, as it is the same as the well-established IS, which was, however, sampling on the rooftop and not via the WAI. As compared to FRIDGE, INSEKT might be more reliable towards warmer temperatures due to the larger sampling volume.

L585-586: Can the authors add the comparison between two different samplers in the appendix?

The comparison is already shown in Figs. 5 – 7, where both samplers were used to collect filters in parallel, and were analyzed with INDA and LINA. We relate this statement now to these figures.

*„Figure 8 includes only data for INDA and LINA obtained from the standard filter holder, as no influence from the two different samplers (standard and HERA) were observed (see Figs. 5 – 7).“*

L588-L593: Tab. 2 -> Table 2. Please refer to ACP guidelines. Does droplet size (2.5 µL vs. 11 µL) play a role here?

Corrected. We believe that the difference does not arise from the different droplet sizes, and state our reasons for this in the text:

*„In addition, the methods use different suspension volumes for the INP detection. However, measurements with INSEKT and FRIDGE at the Jungfraujoch show a good agreement (Lacher et al., 2021), which indicates that the larger uncertainty in the present study was not caused by the different suspension volumes, but rather arises from the larger uncertainty in the sample flow from FRIDGE.“*

T594-595: Which stage(s) and size(s) did UNAM-MOUDI-DFT use to quantify INP concentration?

For this study, particles collected from stages 2 to 7 were used, which we clarify now in the method section 2.3.7:

*„Aerosol particle collection was carried out by an inertial cascade impactor (MOUDI 100R, MSP) which divides the particles according to their aerodynamic diameter in each of its 8 stages (cut sizes: 0.18, 0.32, 0.56, 1.0, 1.8, 3.2, 5.6, and 10.0 µm). For this study, particles impacted on stages 2 to 7 were used.“*

L596-597: Again, SEM images before and after washing would help.

The reviewer is right that this would be very helpful measurements, however, such measurements were not performed yet. Ice nucleation experiments are sometimes performed on washed glass coverslips where the observed average median freezing temperature are close to -36°C, as shown in Córdoba et al. (2021). This means that freezing temperatues observed above -35°C are no caused by the glass coverslips. We will take into account the reviewer suggestion in upcoming studies.

L598: Puy de Dome -> Puy de Dôme.

Corrected.

L602-603: miss -> lose. By saying "impaction", do the authors mean particle impaction on the WAI surface? nanometer-sized what? It would be helpful to perform an estimation of diffusion loss.

We corrected the sentence to:

*„This could be supermicron particles, that are lost by impaction in bends, or might not be sampled especially under high-wind conditions, and nanometer-sized particles that are lost by diffusion.“*

L605: Again, what's the dominant source of INP at Puy de Dôme in October?

We do not know the dominant source of INP, no study on the investigation of the INP at Puy de Dôme in October was performed, as this is technically complex (e.g., using PCVIs downstream of online cloud chambers).

P616-617: Please reword.

We changed the sentence to:

*„This indicates that the discrepancy between rooftop and WAI samples does not only arise from a non-sampling of larger INPs, at least those remnants in liquid suspensions after the first freezing experiment was performed."*

L634: Please reconsider a more informative and precise title.

As there is no title in line 634, we assume that the reviewer means the title of the next section 3.2.1 „Comparison to aerosol particle measurements".

We change the title now to *„Investigation of INP differences using aerosol particle measurements"*

L640: What is "the presence of aerosol particles"?

Changed to:

*„In order to get a better insight into this deviation, the time series of the difference between the INP concentration measurements from the IS (rooftop) and INSEKT (laboratory) is investigated in relation to the wind velocity and the concentration of aerosol particles."*

L643: Is "lognormal difference" appropriate here?

As the measured INP concentration are a strong function of temperature and vary over several order of magnitudes between the represented temperatures of -10 °C, -15 °C, and -20 °C, it was necessary to use a lognormal difference.

L645-646: Please clarify these are number concentrations.

Done.

L647: larger than 0.5 and 1 µm -> between 0.5 – 2.5 µm and 1 – 2.5 µm.

Corrected.

L649: Does the inlet refer to WAI discussed between L604-609? If yes, how come $PM_{2.5}$ are mostly lost when 10 µm particles have a transmission efficiency above 60%?

This is due to the fact that those measurements are derived from the CSU-CFDC, that is operated with an impactor, which we explain in section 2.1.

L666: Please reconsider a more informative title.

We changed the title to *„3.2.2 Comparison of INP concentrations using quartz fiber and polycarbonate filters"*

L671-673: Please reword.

We write now:

*„While both LINA and INDA can analyze particles collected with polycarbonate filters (creation of solution using the washing water), only INDA can analyze quartz fiber filter punches that are immersed in ultra-clean water.“*

L682-683: Shouldn't this conclusion be drawn by comparing the results for INDA and LINA using quartz fiber filters, and polycarbonate filters with 200 nm and 800 nm pores?

This suggested comparison was already presented, in the above:

*"HERA was equipped with polycarbonate filters (200 nm pore diameter), and the standard sampler with the quartz filter. Figure 11 shows results from sampling intervals between the 9th (daytime) to the 11th (daytime) of October."*

An added value of the here presented intercomparison campaign is, that it is also possible to compare to other measurements (of which all offline-methods did not vary the type of filter used during the campaign). Therefore, the presented analysis is an extension of the comparison above which would include only TROPOS instrumentation.

L689: Please specify "the examined size range".

Corrected.

*„They reported that the collection efficiency for polycarbonate filters with 800 nm pore sizes and the flow rates used here ( > 11 LPM) are above 97% for all particles in the examined size range (10 − 412 nm).“*

Figure 1:

Full names of the acronyms should be given in the caption.

Corrected.

L166: within -> with

We write now:

*„…online INP measurements are compared within a time span of 10 minutes…“*

Table1:

Instrument names should be consistent throughout. Please define UNAM-MOUDI-DFT.

We use now „UNAM-MOUDI-DFT" throughout the manuscript, and define the abbreviation.

Figure 2:

It would be helpful to indicate the factors of 2 and 5 ranges for the INP concentration of CSU-CFDC.

For clarity, we would like to keep the figure as it is.

How are the error bars for each instrument calculated? Please elaborate.

In section 2 we described the calculation of the error bars for the CSU-CFDC:

„Statistical significance and confidence intervals for each ambient measurement are determined using the moment-based Z-statistic defined in Krishnamoorthy and Lee (2013).

And SPIN:

„The uncertainty in INP concentration for SPIN represents the standard deviation during a 10-minute sampling period.“

And PINE:

„The uncertainty for the INP concentration is 20%, which is an upper estimate from the uncertainties of the determination of the optical detection volume.“

The INP concentrations exceed 20 #/L at -30 °C. Do the authors have an explanation for such a high INP concentration?

This is an interesting question. One possibility is that, in general, the Puy de Dôme experiences elevated INP concentrations as it is impacted by boundary layer air. Recently, a PINE was installed for permanent operation at the site, such that a better insight into the variability in INP concentration will be investigated.

L490: Please keep the same order of instruments in the caption and legend.

Corrected.

L491: The time resolution of CSU-CFDC doesn't seem like 1 minute.

The time resolution of processed data is 1 minute, see the figure below where we show the same plot for 11:00 – 11:30:

[Figure]

Can the authors provide the full-time series in the appendix?

The INP concentration measurements from the full campaign will be presented in a separate publication (Freney et al., in preparation), thus we would not like to show it in this study.

Figure 3 and Table 2:

Why do the three online instruments have so few inter-comparable data points with a much higher time resolution?

Please see our answer to your question above.

Is there a specific reason for the authors to choose CSU-CFDC and INSEKT as the baselines/references for online and offline INP measurement techniques? Do the intermediate measured values by these two instruments play a role in reference instrumentation selection? L493-496 and L582-584 partly address the concern.

We specify our reasons for choosing CSU-CFDC and INSEKT in sections 3.1 and 3.2, respectively:

*„..the results from all intercomparison experiments are investigated using the CSU-CFDC as a reference instrument, given its long history of operation and good characterization."*

*„This method was chosen since filter collection for INSEKT was performed in the laboratory at the WAI inlet, similar to that for most of the other methods, and since it covers a large temperature range (approximately from -8 °C to -25°C) of INP measurement."*

The information in Table 2 can be merged into Fig. 3 and Fig. 8.

For a better overview of the results, we would like to keep the table as it is.

Figure 4:

Inconsistent usage of instrument name in the caption and legend.

Corrected.

The information has been included in Figs. 5 - 7?

We do not understand the question about which information should be included in Figs. 5 - 7.

Figures 5 - 7:

The marker colors for online instruments are hard to distinguish. Please consider changing marker shapes for different online instruments.

For a better distinction from the offline instruments, we would like to keep one symbol color for the online instruments.

Is it better to keep one representative panel, and move the other to the appendix?

For visualizing the agreement and the discrepancies between the methods, that vary with sampling periods, we would like to present all the panels. By showing only one panel, it might give a biased view on the results. We acknowledge that Figure 4 already shows the timeseries of the INP concentration at key temperatures, however, we would like to show all measurements at all temperatures, and in addition, the INP community is most familiar with the interpretation of freezing spectra.

Please include filter information.

We include now the information in the figure captions.

L577: missing "of".

Corrected.

Figure 8:

Inconsistent usage of instrument name in the caption and y-label.

Corrected.

Figure 9:

The symbols are hard to read.

Due to space limitations, we would like to keep the symbol size.

Figure 10:

Please change the color and label color of the right y-axis in panel d.

We do not see the need to change the label color of the right y-axis; due to the markers and the y-axis label, it is clearly defined which y-label refers to the markers.

L662: Please reword the caption.

We do not see the need to change the caption.

References

Beall, C. M., Lucero, D., Hill, T. C., DeMott, P. J., Stokes, M. D., and Prather, K. A.: Best practices for precipitation sample storage for offline studies of ice nucleation, Atmospheric Meas. Tech. Discuss., 1–20, https://doi.org/10.5194/amt-2020-183, 2020.

Boose, Y., Kanji, Z. A., Kohn, M., Sierau, B., Zipori, A., Crawford, I., Lloyd, G., Bukowiecki, N., Herrmann, E., Kupiszewski, P., Steinbacher, M., and Lohmann, U.: Ice Nucleating Particle Measurements at 241 K during Winter Months at 3580 m MSL in the Swiss Alps, J. Atmos. Sci., 73, 2203-2228, 10.1175/JAS-D-15-0236.1, 2016a.

Córdoba, F., Ramírez-Romero, C., Cabrera, D., Raga, G. B., Miranda, J., Alvarez-Ospina, H., Rosas, D., Figueroa, B., Kim, J. S., Yakobi-Hancock, J., Amador, T., Gutierrez, W., García, M., Bertram, A. K., Baumgardner, D., and Ladino, L. A.: Measurement report: Ice nucleating abilities of biomass burning, African dust, and sea spray aerosol particles over the Yucatán Peninsula, Atmos. Chem. Phys., 21, 4453-4470, 10.5194/acp-21-4453-2021, 2021.

Gute, E., Lacher, L., Kanji, Z. A., Kohl, R., Curtius, J., Weber, D., Bingemer, H., Clemen, H.-C., Schneider, J., Gysel-Beer, M., Ferguson, S. T., and Abbatt, J. P. D.: Field evaluation of a Portable Fine Particle Concentrator (PFPC) for ice nucleating particle measurements, Aerosol Sci. Tech., 53, 1067-1078, 10.1080/02786826.2019.1626346, 2019.

Wolf, M. J., Zhang, Y., Zawadowicz, M. A., Goodell, M., Froyd, K., Freney, E., Sellegri, K., Rösch, M., Cui, T., Winter, M., Lacher, L., Axisa, D., DeMott, P. J., Levin, E. J. T., Gute, E., Abbatt, J., Koss, A., Kroll, J. H., Surratt, J. D., and Cziczo, D. J.: A biogenic secondary organic aerosol source of cirrus ice nucleating particles, Nat. Commun., 11, 4834, 10.1038/s41467-020-18424-6, 2020.

---

## Author Comment (AC2)

**Answer to Reviewer 2**

We thank the referee for commenting our manuscript and giving very valuable suggestions.

The reviewer's comments are in blue, and our answers in black. Sections from the original manuscript are presented in *black italic* and corrections in *red italic*.

Lacher et al, do an extensive intercomparison of INP measurement techniques at the Puy de Dome research station. They generally find that all of the measurement techniques agree reasonably well, especially when accounting for instrumental differences and the sampling location (in front/behind the inlet). The manuscript is extremely well written and shows that the INP community has developed a suite of different INP measurement techniques that are capable of robustly measuring INPs in the field. I recommend the manuscript is published once the comments below are addressed.

General comments:

This is a very technical paper, which makes huge efforts to do a much needed instrument comparison. However, due to its technical nature and since it is primarily an instrument intercomparison, I wonder if it is more appropriate for it to be published in AMT. I know this is up to the authors and editor, but it is something I would consider since it mainly discussing and comparing instrument sampling discrepancies, rather than investigating processes/ atmospheric concentration of INPs.

The reviewer raises a good point, and we believe that the manuscript is suitable to be published in both journals, ACP and AMT. The reason why we submitted it to ACP is that the discussion about the comparablity is specifically taking into account that we sample ambient air, and interpret the sampling conditions such as available aerosol particle properties and meteorology that might impact the intercomparison. Please note that other intercomparison studies (e.g., DeMott et al., 2017; Burkert-Kohn et al., 2017) are also published in ACP. Moreover, we plan to submit a companion manuscript about the interpretation of the INP concentration to ACP as well. Thus, we would like to submit our manuscript to ACP, but leave the decision to the editor.

As I was reading, I often found myself wondering why the instrument intercomparison did not do more to ensure that everything was the same across measurement platforms (e.g. same filter water distributed across DFTs, same set points on the CFDCs). Then I realized that one of the strengths of the studies is that the comparison was conducted using the native sampling format of each measurement technique. I think this point could be strengthened/empahsized, as it is a very nice conclusion showing that all of the tested techniques give representative INP concentrations and are equally useable in the field.

Thank you for pointing this out. We now add the following sentences to the end of the introduction section to highlight this important point:

„*During PICNIC, seven offline techniques and three online instruments were compared over 14 days in October 2018. The aim here was to test the measurement techniques against each other in their original operation configuration, as each of them are well-established methods and were already used in several campaigns, and we wanted to create a link between these activities without changing measurement protocols*."

At the beginning of section 4 Summary and conclusion:

*„During the PICNIC campaign in October 2018, a suite of online and offline INP measurement techniques were operated simultaneously to compare the temperature-dependent INP concentrations relevant to the formation of mixed-phase clouds. The methods were deployed in their typical operation configuration without equalizing measurement setups.“*

And at the end of section 4

*„With regard to required precision of INP measurements to be included in models, the results from our study greatly demonstrate that the methods as used in their original configuration agreed overall well within a factor of 5.“*

It is not immediately clear if all the INP concentrations are reported in stdL$^{-1}$ or not (although the figure units are stdL-1). Perhaps make it clear how this is reported especially when for some of the filter techniques (ie IS, INSEKT) a mass flow is explicitly mentioned.

We added now at the beginning of the online and offline sections, as a general remark, that all INP concentration measurements are given in standard liter.

*„Three different online INP instruments were operated behind the WAI in parallel for several hours per day. INP concentrations were determined for single particles activating in a temperature range between ~ -20 °C and  30 °C, in the condensation/immersion freezing mode (via controlling processing relative humidity). All INP concentrations are referenced to standard liters sampled.“*

*„For offline INP analysis, aerosol particles were collected simultaneously with the different sampling setups during 8-hour intervals. ). All INP concentrations are given with reference to standard liters sampled.“*

At a more critical level, sometimes the main messages of the paper were not very clear. First, there is a lot of discussion about why the instruments may not agree completely. I appreciate the authors taking the time to discuss these uncertainties/factors. However, all of the discussion is quite speculative and some calculations could determine if the proposed factors are the culprit or not. Second, is such a long discussion warranted when the measurement techniques agree so well i.e. majority agreement within a factor 5 or better. Along the same lines, at what point is the agreement high enough that we no longer need to do intercomparisons? Have we reached this agreement level? How much of an influence would a factor of 5 agreement have on the development of parametrizations and modelling results? Lastly, if we don't expect to get closer than a factor of 5, do we really need more field intercomparisons? Wouldn't it make more sense to do careful lab intercomparisons where we can adjust our techniques to have better agreement in a controlled setting?

Thank you for this remark. We hope that with addressing the comments from you and the other reviewer that generally the quality of the manuscript improved, and that its main message is now more clear.

With regard to the in-detailed discussion of potential discrepancies, the aim here was not only to present our measurement results but also to point out certain factors that can be a source of disagreement. However, as the reviewer points out, the discussion is speculative, as with the experimental setup used, we cannot pinpoint reasons for disagreement. Along this line, although

an agreement within a factor 5 might good enough for model representation of INP concentrations, it still is a relatively large number. Besides, we argue that the sensitivity of models to the INP number concentration is still not fully understood, and this might change when modeling different cloud types in different environments. Moreover, it is vital to investigate the reasons for this deviation, as they might have a greater impact sampling a different aerosol and INP population. E.g., the observed discrepancy between the samples collected on rooftop and behind the WAI might be greater when, e.g., sampling a dust-dominated INP population where more large particles are present that might lead to particle break-up in solution. Thus, as we do not know the factors yet that lead to discrepancies, and as setups and operation procedures of individual methods might be changed for improvement, frequent intercomparisons of methods are recommended.

While laboratory intercomparison studies are very useful to limit factors that can lead to disagreement between methods, intercomparisons of instruments that are operated in field campaigns are required, as some factors cannot be easily simulated in the laboratory, as e.g., the impact of windspeed on the transmission efficiency of particles, or simply the usage of a whole air inlet. In the lab, agreement often depends on used aerosol samples, while in the field a mixture of natural aerosol can lead to different results. Again, as we do not fully understand why there are discrepancies between instruments, we need to compare with well-known (laboratory) and ambient (field) aerosols. As most of the instruments are used for field applications, it is certainly needed to intercompare under realistic sampling conditions.

We are not aware of calculations besides the particle transmission efficiency for the WAI that could be used for determining factors leading to disagreement between the instruments.

Minor comments:

Abstract: Consider adding a discussion about the importance of INPs as this is geared for ACP, if AMT is the end journal then it is fine as is.

This is a very good suggestion. We changed the start of the abstract to:

„*The formation of ice crystals in clouds is initiated by specific aerosol particles, termed ice-nucleating particles (INPs). Only a tiny fraction of all aerosol particles are INPs and their concentration over the relevant temperature range for mixed-phase clouds (< -38 °C) covers up to ten orders of magnitude, providing a challenge for contemporary INP measurement techniques.*"

Line 155-156: Is there a reference for these reported transmission efficiencies?

Yes, we used calculations to report these numbers, which we clarify now and include the respective references:

„*Moreover, the transmission efficiency of the WAI is dependent on wind speed. Calculations show that at values of 7 (10) m s$^{-1}$, 93% (84%) of the particles with a diameter of 10 µm are entering the inlet (Hangal and Willeke, 1990; Baron and Willeke, 2002).*"

Line 181-193: This is a bit unclear. So the concentrator was only used occasionally but how does this work when comparing between PINE, which fills for a fixed interval and then expands while the CFDCs measure continuously? Was the CFDC data excluded when PINE was filling as there would

be a different flow ratio in the concentrator when all three instruments sampled? Do you expect this to be a potential problem/ lead to a change in the concentration factors ie change in particle size distribution?

The PINE inlet flow rate is constant, during the expansion the inlet flow is maintained by a bypass flow. Thus, the flow ratios of the PFPC was kept constant. We add to the PINE method's section now:

*„The flush mode is followed by the expansion mode when a valve upstream of the chamber is closed while the volumetric flow out of the chamber is set to a constant value of 3 LPM. Please note that the inlet flow rate during the expansion is maintained by a bypass flow which is the same as the flush flow rate, such that no change in the sampling flow at the WAI occurs."*

Role of impactors: Does it matter that SPIN uses one impactor and CSU-CFDC is using two? The impactor cut size is distribution based so would this lead to more lager particles making it into SPIN than the CSU-CFDC? Of course this does not seem to be the case based on the results but it could be something to mention if this another source of uncertainty. I see this is now discussed a bit in the results. Perhaps it would be nice to have a table with the online instruments and their associated set up e.g., impactors, set conditions etc.

We agree that a small difference due to the use of two impactors might occur, something the first reviewer pointed out as well. Hovewer, this would lead to more larger particles entering SPIN and thus likely to a higher INP concentration as compared to CSU-CFDC, which is not observed. For clarification, we add now to the end of section 3.1 (intercomparison of online instruments) the following sentences to address the difference in the impactors used:

*„It should be noted that differences between the online instruments might arise from the difference in impactors. CSU-CFDC is operated with two single-jet 2.5 µm impactors, while SPIN is using only one, and PINE is operated without an impactor and thus has a 50% aerodynamic size-cut at 4 µm due to the loss of particles in its inlet."*

More, we appreciate the idea of adding a setup table similar to the table provided for the offline methods, and included the following table in section 2.2

*Table 1: Specifications of the online instruments.*

| Name | CSU-CFDC | SPIN | PINE |
|---|---|---|---|
| inlet | WAI / PFPC* | WAI / PFPC* | WAI / PFPC* |
| impactor | two impactors with 2.5 µm size-cut | one impactor with 2.5 µm size-cut | no impactor; size-cut 4 µm (Möhler et al., 2021) |
| temperature and RH$_{water}$ uncertainty | ± 0.5°C and 2.4% | ± 0.5°C and 2.5% | ± 1 °C |
| residence time | 5 s | 10 s | < 33 s |
| supersaturation | 106.5% RH$_{water}$ | 102.8% RH$_{water}$ | > 100% RH$_{water}$ |
| ice threshold | 4 µm | 5 µm | automated |

\* online instrument sampled always at the same inlet

Section 2.2.2- Is it concerning that the RHw in spin was ~3 % lower than in the CSU-CFDC yet the ice crystal detection was one micron higher? It might be worth mentioning that the longer growth

(residence) time in SPIN would ensure that at this supersaturation, the ice crystals would reach a 5 micron size even with the lower RHw than in the CSU-CFDC if this is the case. This is also discussed later but not actually calculated. Consider doing this calculation.

Indeed, due to the longer residence time in SPIN, ice crystals are growing to larger sizes as the CSU-CFDC even though SPIN is operated at a lower superaturation. To make this point clear, we add to section 2.2.2 now:

*„Due to the sigmoidal shape of the impactor's size cut, OPC counts larger than 5 μm in diameter were considered as activated INPs. Although SPIN is operated at a lower supersaturation as compared to the CSU-CFDC, the ice crystal have a longer residence time (10 seconds) such that they grow to sizes larger than 5 μm."*

Line 258-259: Why does the limit of detection double behind the concentrator? Aren't the background counts from the wall the same regardless of the concentrator? I this just due to having more "air" going into the instrument? This is not immediately clear, consider explaining this a bit better.

Likely our statement made was not clear here. Indeed, the limit of the detection of SPIN behind the concentrator is lower as compared to sampling not at the concentrator, as more sample is analyzed while the background counts are the same. We correct the statement such that this is clear now.

*„The limit of detection of SPIN sampling at the concentration is lower (~ 0.6 INP L$^{-1}$) as compared to not sampling at the concentrator  (~ 6 INP L$^{-1}$), as more sampled air is analyzed, while the ice background counts remain the same."*

Section 2.2.3 -

Firstly, does this mean that PINE can only measure between -19 and -13 based on the start temperature of -13 and only a 6 degree cooling? This is not immediately clear. It might be nice to add a table with the temperatures that the comparisons were conducted at (see previous comment about impactors).

No, this is not the case, we operate PINE at temperatures *below* the frost point temperature of ~ -13 °C. Indeed, we operate PINE typically at temperatures colder than the frost point temperature. That means that the sampled air is too humid, thus we have an ice layer on the inner chamber walls, that is, however, not impacting our sampling conditions in terms of background from internally formed ice crystals. As the total water content at temperatures of below -13 °C is smaller as compared to warmer temperatures, no de-icing of the chamber walls within a shorter sampling period of a few days is needed (for a long-term operation with PINE, the chamber is de-iced frequently).

The INP concentrations were compared between -20 °C and -30 °C, as written in section 2.2

*„INP concentrations were determined for single particles activating in a temperature range between ~ -20 °C and  30 °C…"*

and indicated in figures 2 and 3. Thus we believe that this is redundant information for the new table 1, that we wanted to keep similar to table 2 (offline methods) and does not contain this information likewise.

Second, during the continuous expansion, aerosol particles are removed, therefore is there some sort of correction for the decrease in effective volume in the chamber? I ask as depending at the ambient pressure (I guess around 850 hPa) a 300 hPa decrease is ~35 % loss of aerosol during the expansion. This would suggest that depending on when in the cycle the set temperature is reached, a fraction of the aerosol are already removed (albeit still less than the factor of 2 threshold). Please add information about this correction if it is performed or necessary.

This is an interesting question. The intention in PINE is to analyze all aerosol particles within the volume of the chamber, which is achieved by guiding the expansion flow containing the aerosol particles and formed ice crystals over the OPC attached between the chamber and the pump. Thus, all aerosol particles that are „lossed" during the pressure decrease are measured. Indeed, the temperature that is assigned as the nucleation temperature is the minimum temperature reached at the end of the expansion (Möhler et al., 2021), meaning that a fraction of the particles were not exposed to this minimum temperature. However, we do not apply a correction factor for this, but might consider this for future studies.

Third, does the lack of impactor here also mean larger particles are expected to enter PINE than the other two online measurements since a cut size is distribution based?

Yes, this is correct, see our answers to your comment above.

Table 1: Consider adding the total volume of the air sampled for each filter technique as well as the minimum INP detection limit for each instrumental technique in the table. I know that this is corrected for, but I still think it is nice to see since the range in INP concentration covered appears to vary quite a bit.

Thank you for this suggestion. We decided to include now the limit of detection to table 2 (formerly table 1).

| | Name | FRIDGE | INSEKT | INDA* | IS | LINA* | LINDA | UNAM-MOUDI-DFT |
|---|---|---|---|---|---|---|---|---|
| filter collection | location | WAI | WAI | WAI | rooftop | see INDA | rooftop | WAI |
| | time interval | 8 hours (night), 4 hours (day) | 8 hours | 8 hours | 8 hours | same as INDA | 8 hours | 8 hours |
| | substrate | 47 mm PTFE fluoropore membrane filter, 220 nm pore size | 47 mm polycarbonate filters, 200 nm pore size | 47 mm polycarbonate filters, 200 nm and 800 nm pore size; 47 mm quartz fiber filters | 47 mm polycarbonate filters, 200 nm pore size | same as INDA | 15 cm quarz fiber filters | hydrophobic glass coverslips |
| | filter holder | custom-built semi-automated multi-filter sampling device | standard | standard, HERA | open-faced sterile Nalgene sampling heads | same as INDA | high volume sampler | MOUDI cascade impactor |
| | flow | 4.8 LPM | 11 LPM | standard: 12 - 37 LPM HERA: 15-41 LPM | 13.5 LPM | same as INDA | 500 LPM | 30 LPM |
| | limit of detection (L$^{-1}$) | 4.3E-04 (8 hours) | 1.90E-04 | standard: 1.7E-04 - 5.6E-05 HERA: 1.4E-04 - 5.1E-05 | 1.5E-04 | same as INDA | 4.2E-06 | 6.9E-05 |
| | filter storage | partly unfrozen | frozen | frozen | frozen | same as INDA | frozen | refridgerated |
| analysis | liquid volumes | 2.5 µl droplets | 50 µl suspension | 50 µl suspension | 50 µl suspension | 1 µl droplets | 200 µl suspension | 100 µm droplets |
| | cooling rate | 1 °C min$^{-1}$ | 0.3 °C min$^{-1}$ | 1 °C min$^{-1}$ | 0.3 °C min$^{-1}$ | 1 °C min$^{-1}$ | 0.3 °C min$^{-1}$ | 10°C min$^{-1}$ |

* INDA and LINA use the same collected filter

Line 343-344: Consider adding a ref where cold transportation is discussed e.g. Beall et al., (2020)

We include now the following reference:

*„The samples were not actively cooled during transport, however, given the relatively short travel time of ~ 8 hours to the laboratory in Frankfurt, we do not consider that this impacts the results, but cannot exclude it for certain (Beall et al., 2020)."*

Line 369: You could add Sarah Grawe's new paper about HERA if you wanted (Grawe et al., 2023)

This is a good point, we include Grawe et al. (2023) now.

Section 3.2-

Here it would be worth mentioning the probability of detecting INPs at the warmer temperatures due to the limited sampling volumes used. It looks like LINDA and IS, which are also two of the methods using the equivalent of the most volume of air sampled, are the highest.

Agreed. We add to the beginning of section 3.2 now:

*„Due to the difference in sampled volume and thus detection limit (see table 2), the probability to detect very rare INPs at temperatures above ~ -10°C varies amongst instruments."*

Line 456-458: Again it would be worth citing the importance of transporting samples frozen or at least the impact of warmer temperature transport (e.g. Beall et al., 2020).

We add now to the description of the transportation of samples for analysis with UNAM-MOUDI-DFT:

*„We note that storing samples for a longer transportation time might impact the INP concentration (e.g., Beall et al., 2020). Although we did our best to keep the samples below 0°C by transporting them in a freezer with ice packs, it is very likely that the samples may have experienced temperatures slightly above 0°C right before reaching their final destination."*

Line 466: What are the impacts of such a high cooling rate? Previous studies have looked into the impact of cooling rate (e.g. Budke and Koop, 2015) and have seen a difference due to the stochastic nature of ice nucleation. This should also be discussed more in the results and not just as a conclusion. See point again below.

We agree that such a conclusion should be discussed in the results, and add now to the results section about UNAM-MOUDI-DFT:

*„As shown in Fig. 8b, the UNAM-MOUDI-DFT tends to measure higher INP concentrations compared to INSEKT. This bias may be coming from the method used to capture the particles. While for the INSEKT samples Nuclepore filters were used, in the MOUDI-DFT particles were impacted on glass coverslips. A possible explanation is that not all particles are released from the Nuclepore filters. If so, this may relate to the aerosols sampled at Puy de Dome, as this bias was not seen in some prior comparisons (e.g., Mason et al., 2015). Moreover, the UNAM-MOUDI-DFT is the method using the fastest cooling rates of 10 °C per minute, such that an effect of a time dependency of ice nucleation might have impacted the results (e.g., Hoose and Möhler, 2012; Budke and Koop, 2015). However, this would have lead to an underestimation of INP concentration, such that we conclude that the ambient INP concentration are not considerably controlled by stochastic variation, or that other instrumental properties of sample colletion and analysis with UNAM-MOUDI-DFT are dominant."*

Line 495: Even though it is well known that the CSU-CFDC is a well-known and established measurement system, please add some references that attest to its established reputation.

We add now a selection of campaigns including measurements with CSU-CFDC:

*„To identify potential systematic deviations between the three instruments, the results from all intercomparison experiments are investigated using CSU-CFDC as a reference instrument, given its long history of operation and good characterization. CSU-CFDC has been used extensively in laboratory intercomparisons (e.g., DeMott et al., 2011; Hiranuma et al., 2015; DeMott et al., 2018) and in a large number of field measurement studies in surface- and aircraft-based campaigns (e.g., within the last five years, McCluskey et al., 2018; Cornwell et al., 2019; Hiranuma et al., 2019, Kanji et al., 2019; Levin et al., 2019; Schill et al., 2020; Barry et al., 2021; Knopf et al., 2021; Twohy et al., 2021) over a period of more than 25 years."*

Editorial comments:

Line 10: please switch to listing coldest temperature first followed by the warmer temperature when giving a T range. This should be consistent throughout the entire manuscript.

Throughout the manuscript we are listing warmer temperatures first, thus this is consistent and we would like to keep this ordering.

Line 38: Although this is a complete list, you could add studies that show a relationship between INP concentrations and cloud phase e.g. (Creamean et al., 2022; Carlsen and David, 2022; Sze et al., 2023)

Thank you for this suggestion, we added the proposed publications to the manuscript.

*„The presence of INPs is important for the formation and further development of clouds since they can determine cloud phase (e.g., by a rapid cloud glaciation and associated dissipation effect; Campbell and Shiobara, 2008; Murray et al., 2012; Paukert and Hoose, 2014; Kalesse et al., 2016; Desai et al., 2019; Murray and Liu, 2022; Carlsen and David, 2022; Creamean et al., 2022; Sze et al., 2023)..."*

Line 74-78: It would be nice to add a reference for this.

We added Rogers et al. (2001) now.

*„Typically, online instruments, such as continuous flow diffusion chambers (CFDCs), limit the aerosol sampling to size to diameters below ~ 3 μm (e.g., Rogers et al., 2001),..."*

Line 149:program-> programs.

Corrected.

Figure 1: Please have T range go from cold to warm.

See our answer above.

Line 308: remove the additional "periods".

Done.

Eq. 1: is it appropriate to call it a $V_{sol}$ as you later refer to it as the droplets containing the suspension? Consider calling it $V_{sus}$ or similar for suspension, but this is just a semantics thing.

We agree with the reviewer and call it now $V_{sus}$:

„*The calculation thereby considers the volume of water used to extract the sample (*suspension; $V_{sus}$*)…*"

We corrected it to (Hiranuma et al., 2015).

Line 347: Consider adding that the filter had been exposed or was a sample containing filter or something similar.

We add now:

"*Before starting a measurement, a filter* containing the sampled aerosol *was placed in a sterile Eppendorf tube, which was filled with 5 mL of ultrapure water (Rotipuran ultra, Carl Roth).*"

Line 495-496: It's a bit unclear here what is meant by the true INP concentration? Does this lead to an undercounting or overcounting? My guess would be undercounting, I would specify this.

We corrected the sentence to:

„*However, it should be noted that also the CSU-CFDC might* not measure the total ambient *INP concentration, due to aerosol lamina properties and size cuts, which will be discussed below in more detail* and that can lead to an underestimation of the INP concentration.*"

Corrected.

We re-organized the explanation of the different factors that can impact the determination of the INP concentration in SPIN:

„*Previous studies have found that the aerosol particles in at least some CFDCs are likely spreading beyond the lamina, such that not 100% of particles are in the lamina where they are exposed to the targeted supersaturation condition (DeMott et al., 2015; Garimella et al., 2017; Wolf et al., 2019). The issue of lamina spreading is likely variable and depends on the CFDC geometry, the flow conditions, and the temperature gradients between the walls, which is creating the supersaturation;*

*ultimately this may be an issue with how the central lamina is introduced to the chamber, and the thermal gradients existing there. Aerosol spreading causes aerosol particles to experience lower supersaturations than the target supersaturation, resulting in either a non-activation into cloud droplets and ice crystals (immersion freezing mode), or an activation into ice crystals that are not growing to sizes within the residence time in the chamber to be detected by the OPC (above the ice threshold). SPIN was operated at a lower supersaturation (2.8 ± 1.9%) as compared to CSU-CFDC (6.5 ± 1.4%). Thus, it is expected that SPIN underestimates INP concentration, by up to a factor of 10 (Garimella et al,, 2017; Wolf et al., 2020)."*

*Moreover, SPIN also used a larger ice threshold in the OPC of 5 μm, against 4 μm from CSU-CFDC, which has been found to impact INP concentration measurements (Jones et al., 2011). Thus, it is possible that due to a larger ice threshold size, fewer particles in SPIN were encapsulated in the intended conditions, and were less likely to reach the critical size threshold."*

Line 514: comma before which.

Corrected.

Line 520-527: Here the discussion of spreading is well discussed. I would consider reordering and only mention these issues once, like done in this section.

See our answer above.

Line 745-747: Please make sure this is also discussed in the main text as it is an interesting point to raise. Does it make sense to consider longer time scales for the importance of time-dependence at these warm temperatures i.e. maybe you need longer time scales like 5-10s of minutes for the stochasticity to matter e.g. Budke and Koop, (2015) who needed a very low freezing rate to observe a big difference (.1 K/min)?

We added a sentence in the main text, see our answer to your question before.

The reviewer is right that longer time scales might be needed. However, the intention of this study was not to investigate a potential time dependency of ice nucleation, but to compare methods used for ambient measurements, thus we would not like to add a statement or suggestion here for future experiments on a potential time-dependence.

References:

[revised manuscript text omitted]

---

## Author Response (AR2)

**Answer to the editor's comments**

We thank the editor and the editorial team for their careful thoughts about the missing criteria of the manuscript to be published in ACP. We have now added discussions and statements about the atmospheric implication of our intercomparison campaign. The editor's comments are in blue, and our answers are in black. Sections from the original manuscript are presented are in *black italic* and corrections in *red italic*.

Authors,

Thank you for your revised manuscript and for your very detailed responses to the two sets of referee comments. I also apologize for the extra delay in my response. I needed to discuss the decision with the editorial leadership to make a final decision about the placement of the manuscript. The manuscript is very nice and provides an important and very useful ambient intercomparison of various INP sampling and measurement techniques, even if it is very technical. You have done a good job revising the manuscript based on the feedback, and it is generally close to being ready for acceptance. The point referee #2 introduced about whether ACP or AMT would be a better fit is an important one, and what I needed to discuss. Author guidelines for ACP research articles are now nicely summarized here (with more detail at the web address/link below):

"Implications: Discuss what the results mean for our understanding of the state and/or behaviour of the atmosphere and climate, which is the main requirement for publication in ACP. The editor's acceptance/rejection decision will be strongly guided by this component of the concluding section."
https://www.atmospheric-chemistry-and-physics.net/policies/guidelines_for_authors.html

While the manuscript itself is essentially ready, the editorial team and I don't think the text yet adequately fits the requirements for publication in ACP. In particular, the abstract and conclusions need edits to make them closer to these requirements. Had I thought through these points before review, I would have suggested moving the manuscript to AMT at that point. That would require a new round of review, however, and I don't want to require that of your manuscript at this stage. So I suggest that we proceed toward publication in ACP, but following an additional round of edits to the abstract and conclusions. This final round of revision can proceed through what hopefully should be a very quick round of review only by me, and it will not require a second review by anonymous peers.

Please see the new requirements for publication, linked above. In particular, your abstract is currently at ~475 words, which is well above the new maximum of 250 words. The guide provides a very useful template of components to include, with point 7 (the importance and implications of the results) as a critically missing component for the present manuscript version. I suggest that you add some discussion of the broader atmospheric implications of the study within both the abstract and conclusions, but hold to abstract word limits, as now posted.

More specifically, the last section of your manuscript is still largely a summary of the differences between the different techniques and the technical basis and implications, which is not yet sufficient for an ACP research article. This is where the article "must convincingly demonstrate important implications for our understanding of the state and behaviour of the atmosphere and climate" and not just the implications for how measurements can best be performed or interpreted. The implications of the study are currently expressed primarily in the single sentence added during revision (Lines 836-837: "With the regard to ...").

While you will not be required to address all of these suggestions, it may be useful to address some of these points. For example:
• What does the stated factor of 5 mean for our understanding of the atmosphere?

• Can you be more detailed and specific in your discussion about how this factor of 5 would impact atmospheric models?

• Clearly state what the novelty of the current study is. How do your conclusions add to or alter conclusions from previous reports and the broader, existing understanding of atmospheric INP and/or atmospheric processes?

• The final two sentences of the conclusions ("Ambient INP intercomparison campaigns …") does not provide a powerful ending conclusion, especially given that these summary statements have been broadly known. For example, you might consider specific ways in which you would expect your results to constrain models or improve atmospheric understanding with respect to the needs and questions presented e.g. in publications like these:

o Burrows et al., 2022: https://agupubs.onlinelibrary.wiley.com/doi/full/10.1029/2021RG000745

o Coluzza et al., 2017: https://www.mdpi.com/2073-4433/8/8/138

o Zamin et al., 2017: https://journals.ametsoc.org/view/journals/amsm/58/1/amsmonographs-d-16-0006.1.xml

o Ervens et al., 2011: https://agupubs.onlinelibrary.wiley.com/doi/full/10.1029/2011JD015729

I am happy to answer further questions via email, if you like. Otherwise, I look forward to looking at your revised manuscript.

Best regards,

Alex Huffman

In order to include a statement about the importance of the atmospheric implication of our study, we extended the abstract by highlighting the need for ambient intercomparison studies for a qualified embedding of the INP variables in models. At the same time, to limit the abstract to 250 words, we shortened the abstract by excluding details about the INP methods and specific results about their intercomparability.

[revised manuscript text omitted]

---

## Author Response (AR3)

**Answer to the editor's comments**

We thank the editor for the grammatical corrections in abstract. The editor's comments are in blue, and our answers are in black. Sections from the original manuscript are presented are in *black italic* and corrections in *red italic*.

Authors,

Thank you for revising the abstract and conclusions along the lines of what I suggested in the previous decision. I know how challenging it can be to cut valuable text out of an already finished abstract. I'm happy with your additions and for the manuscript to proceed to publication.

As I read your revisions, I made grammatical edits to two sentences in the abstract.
- "However, for a qualified implementation of INPs in models, measurement techniques able to accurately detect the temperature-dependent INP concentration are needed."
- "Although a variety of different measurement principles were used, the majority of the data show INP concentrations within a factor of 5 of one another, ...."

You can consider whether these changes are helpful or not. Otherwise, please proceed to upload your final documents for typesetting, as to be guided by the Copernicus staff.

Best regards, and Happy New Year.

Alex Huffman

We follow the suggestions of the editor and make the grammatical corrections in the abstract:

*Ice crystal formation in mixed-phase clouds is initiated by specific aerosol particles, termed ice-nucleating particles (INPs). Only a tiny fraction of all aerosol particles are INPs, providing a challenge for contemporary INP measurement techniques. Models have shown that the presence of INPs in clouds can impact their radiative properties and induce precipitation formation. However, for a qualified implementation of INPs in models, measurement techniques able to accurately detect the temperature-dependent INP concentration are needed. Here we present measurements of INP concentrations in ambient air under conditions relevant to mixed-phase clouds from a total of ten INP methods over two weeks in October 2018 at the Puy de Dôme observatory in central France. A special focus in this intercomparison campaign was placed on having overlapping sampling periods. Although a variety of different measurement principles were used, the majority of the data show INP concentrations within a factor of 5 of one another, demonstrating the suitability of the instruments to derive model-relevant INP data.*
*Lower values of comparability are likely due to instrument-specific features such as aerosol lamina spreading in continuous-flow diffusion chambers, demonstrating the need to account for such phenomena when interpreting INP concentration data from online instruments. Moreover, consistently higher INP concentrations were observed from aerosol filters collected on the rooftop at the Puy de Dôme station without the use of an aerosol inlet.*